# Changing Snow Water Storage in Natural Snow Reservoirs

Christina Marie Aragon[1] and David F. Hill[2]

[1]Oregon State University, Water Resources Engineering, Corvallis, OR, 97331, USA
[2]Oregon State University, Civil and Construction Engineering, Corvallis, OR, 97331, USA

**Correspondence:** Christina Marie Aragon (aragonch@oregonstate.edu)

**Abstract.** This work introduces a novel snow metric, snow water storage (SwS), defined as the integrated area under the snow water equivalent (SWE) curve [units: length-time, e.g. meter$^3$-days]. Unlike other widely-used snow metrics that capture snow variables at a single point in time (e.g. maximum SWE) or describe temporal snow characteristics (e.g. length of snow season), SwS is applicable at numerous spatial and temporal scales. This flexibility in the SwS metric enables us to characterize the inherent reservoir function of snowpacks and quantify how this function has changed in recent decades. In this research, changes in the SwS metric are evaluated at point, gridded and aggregated scales across the conterminous United States (hereafter US), with a particular focus on 16 mountainous EPA Level III Ecoregions (ER3s). These ER3s account for 72% of the annual SwS (SwS$_A$) in the US, despite these ER3s only covering 16% of the US land area. Since 1982, spatially variable changes in SwS$_A$ are observed across the US with notable decreasing SwS$_A$ trends in the western US and in the 16 mountainous ER3s. All mountainous ER3 (except for the the Northern Highlands in New England) exhibit decreasing trends in SwS$_A$ resulting in a 22% overall decline in SwS$_A$ across mountainous ER3s. The peak monthly SwS (SwS$_M$) occurs in March at all spatial scales, while the greatest percent loss of SwS$_M$ occurs early in the snow season, particularly in November. Unsurprisingly, the highest elevations contribute most to SwS$_A$ in all mountain ranges, but the specific elevations that have experienced loss or gain in SwS$_A$ over the 39-year study period vary between mountain ranges. Comparisons of SwS with other snow metrics underscore the utility of SwS, providing insights into the natural reservoir function of snowpacks, irrespective of SWE curve variability or type (e.g. ephemeral, mountain, permanent). As we anticipate a future marked by increased climate variability and greater variability in mountain snowpacks, the spatial and temporal flexibility of snow metrics such as SwS may become increasingly valuable for monitoring and predicting snow water resources.

## 1 Introduction

Seasonal snow is a critical resource in mountainous regions and at high latitudes across the United States (US), and many other countries, providing an important ecosystem service by functioning as a natural and spatially-distributed reservoir (Barnett et al., 2005). These snow reservoirs play a key role in the water cycle by storing water during the cool season and releasing water gradually throughout the warm season when human and ecological demand is the highest (Li et al., 2017). The natural reservoir function of snowpacks is at risk due to anthropogenic climate change, which has been shown to decrease snowpack magnitude and persistence while increasing snowpack variability (Siirila-Woodburn et al., 2021; Scalzitti et al., 2016; Sospedra-Alfonso et al., 2015; Morán-Tejeda et al., 2013). The variability of climatic variables that drive snowpack variability, such as

precipitation and temperature, has increased in the recent past and is projected to continue to increase as a result of climate change (Scalzitti et al., 2016; Sospedra-Alfonso et al., 2015; Morán-Tejeda et al., 2013; Pendergrass et al., 2017; Ohmura, 2012). Given the vulnerability of seasonal snow water storage to climate change and the importance of snow-derived water to municipalities, agriculture, ecosystems, and hazard forecasters, it is vital to understand how water storage in our natural snow reservoirs is evolving in the context of a changing climate (Immerzeel et al., 2020; Sturm et al., 2017; Barnett et al., 2005; Li et al., 2017; Siirila-Woodburn et al., 2021).

Snow water equivalent (SWE) is a relevant snowpack characteristic for many water resources applications. SWE is the depth of water obtained upon melting a column of snow. Having an estimate of SWE across a watershed is analogous to knowing the stage (water elevation) in a surface reservoir; it quantifies the amount of water being stored for later use. Other directly measurable snow characteristics largely fall into two categories; they can be temporal snapshots that give us information about snow magnitude at a certain point in time, or they can provide information about snow timing (Nolin et al., 2021). April 1st SWE, snow covered area (SCA), and peak SWE ($SWE_{max}$) are examples of temporal snapshot metrics. Snow metrics that give us information about the timing of snow include snow cover duration (SCD), date of snow onset (DSO) and date of snow disappearance (DSD).

Composite snow metrics, such as the Snow Storage Index (SSI) (Hale et al., 2023) and the Water Tower Index (WTI) (Viviroli et al., 2007), are not directly measurable, but combine information streams in order to relate snowpack to water storage. The SSI indicates the degree to which snowpack delays the timing and magnitude of surface water inputs relative to when it falls as precipitation and the WTI identifies locations where mountain runoff contributes disproportionately to lowland water supplies. Additionally, a global WTI was developed, which ranks all water towers in terms of their water-supplying role and downstream societal and ecological demand (Immerzeel et al., 2020).

A conceptual SWE curve is shown in Figure 1(a). The conceptual SWE curve referenced throughout this paper is for mountain snowpacks and is delineated by three points; the DSO the peak SWE ($SWE_{max}$) and the DSD. SWE accumulation begins at the DSO and continues up to a $SWE_{max}$, which may or may not occur on Apr 1 (northern hemisphere). After $SWE_{max}$, the ablation phase of the snow season begins and the SWE depth declines until it reaches zero at the DSD. The SCD is captured by the width of the SWE curve. Multiple factors can result in systematic changes to the shape of the SWE curve including climate change (Lute et al., 2015), natural land cover change such as wildfire (Gleason et al., 2019) or beetle kill (Pugh and Small, 2012; Boon, 2007; Winkler et al., 2014) and man-made land cover change, such as forest thinning (Krogh et al., 2020; Sun et al., 2022) or logging (Winkler et al., 2005; Troendle and Reuss, 1997).

The shaded regions in the other panels of Figure 1 provide plausible examples of how the SWE curve may have changed from the past to present day. For example, a current SWE curve could be a scaled (reduced) version of a past SWE curve (Figure 1(b)). This would result in a later DSO, a lower $SWE_{max}$, an earlier DSD and a shorter SCD. Changes in SWE curves could also result from a temporal shift in the historic curve (Figure 1(c)). This would not impact $SWE_{max}$ or SCD, but metrics including April 1 SWE, DSO, and DSD would be affected. Figure 1(d) gives yet another example of a theoretical current scenario, compared to a historic one. In this case, the shape of the conceptual SWE curve is changed by repeated accumulation and melt events during the accumulation season. As shown in this graphic, metrics such as DSO, DSD, SCD, $SWE_{max}$, and

April 1st SWE could all remain unchanged but it is clear that the snowpack is different than in the past. Previous literature has quantified increasing ablation during the accumulation period by defining a 'melt fraction' (Musselman et al., 2021) which is the ratio of the melt that occurs during the accumulation phase to the total melt. Their metric helps to identify snowpacks that have considerable variance and vulnerability to warming and rain-on-snow events. Another example of changing snowpack is shown in Figure 1(e). Here, the SCD remains constant due to consistent DSO and DSD, but $SWE_{max}$ decreases in magnitude, resulting in less snow overall. Finally, Figure 1(f) shows a theoretical future in which DSO, DSD, SCD, April 1st SWE and $SWE_{max}$ all remain constant, but it is clear that there is less snow present throughout the season.

The conceptual SWE curve and the above discussion is focused on a mountain snowpack, a snowpack with a distinct period of steady accumulation up to a $SWE_{max}$, followed by a similarly steady ablation season that persists throughout the winter. While mountain snowpacks play a key role in natural water storage, other types of snowpacks also have distinct characteristics and are important to the hydrological cycle. For example, ephemeral snowpacks, snowpacks that tend to be have a lower cold content than mountain snowpacks and come and go throughout the winter (Sturm et al., 1995; Hatchett, 2021), play a role in soil moisture and runoff regimes (Livneh and Badger, 2020; Hamlet and Lettenmaier, 2007). Ephemeral snowpacks tend to experience accumulation and ablation processes nearly in tandem (Liston and Elder, 2006). Alternatively, Greenland and Antarctic ice sheets primarily experience accumulation processes (Liston and Elder, 2006). While metrics such as April 1 SWE, $SWE_{max}$ and SCD do a good job of characterizing mountain snowpacks, they are not as useful at capturing the transient nature of ephemeral snowpacks or the lack of an ablation season on ice sheets.

This work aims to characterize the extent to which snowpacks serve as natural reservoirs and evaluate spatial and temporal changes in snow water storage in a new, integrated way. As we move into a future of increased climate and snowpack variability, we need snow metrics that can capture diverse and dynamic snowpack regimes. As the majority of natural snow water storage in the US occurs in mountainous regions, it is important to understand how the natural reservoir function of snowpacks is changing across individual mountain ranges. When attempting to quantify snow water storage change, it can be difficult to merge the scale at which most in-situ observations are available (at the point/station scale) and the scale at which snowpacks operate as natural reservoirs (at the mountain range scale). This study presents a new spatially and temporally flexible snow metric, snow water storage (SwS), in order to address the reality of changing snowpack regimes and the challenge of spatial variability between snow observations and decision making. This study examines SwS trends in mountain snowpacks by addressing the following research questions: (1) What are the trends in monthly and annual SwS across the US at discrete point, gridded and aggregated scales? (2) What is the role of mountainous ER3s in US SwS and how has this changed in recent decades? (3) How does SwS relate to other common snow metrics?

## 2 Methods

### 2.1 Snow Water Storage Metric (SwS)

SwS quantifies the depth of water stored in snow reservoirs over time and is calculated by integrating the area under the SWE curve:

$$SwS = \int SWE(t)\, dt, \tag{1}$$

where SWE has dimensions of length, and integration occurs over a time period (water year, a given month, etc.) of interest. If daily SWE data are used for this calculation at a given point, SwS will have dimensions of meter-days, or md.

As defined above, SwS is a quantity computed at a single point, e.g. a SNOTEL location. However, the SwS metric can also be aggregated across various spatial scales. There are numerous re-analysis products that provide spatially-distributed SWE information on a regular grid. In this case, SwS can be computed for a horizontal area (say a particular watershed) of interest. In this case, the dimensions of SwS will be $m^3d$.

SwS can also be computed for various integration periods. If the integration is done over the entire water year, this yields annual SwS ($SwS_A$). In the integration is for a particular month, this yields monthly SwS ($SwS_M$). Integrating daily SWE data over a single day produces the daily value of SwS ($SwS_D$), but this is simply is the same as daily SWE.

SwS is the integrated area under the SWE curve, indicating the cumulative meter-days of water that was stored as a snowpack. SwS quantifies the degree to which a snowpack functions as a water storage reservoir. Unlike April 1 SWE, or SWE max, SwS can be applied to mountain, ephemeral or permanent snowpacks. Unlike other storage-related metrics such as SSI and WTI, SwS is directly measurable and does not require the combination of multiple data-streams to calculate. Ultimately, this integrated metric helps us to understand how much water is held in our snow reservoirs and for how long.

### 2.2 Data

Daily observations of SWE were obtained from Natural Resources Conservation Service (NRCS) snow telemetry (SNOTEL) stations (Serreze et al., 1999) and from Cooperator Snow Sensors (COOP). The SNOTEL network provides data at discrete scattered points across the western US and the COOP stations used in this study provide data across California. This study used the 465 stations that have a period of record from at least water year 1982 to water year 2020 with less than 10% of days missing during that period.

This study also uses the University of Arizona SWE (UASWE) dataset (Zeng et al., 2018; Broxton et al., 2019) a daily 4-km gridded dataset that spans the US. The UASWE dataset assimilates SWE and snow depth observations into an empirical temperature index snow model that is forced with PRISM temperature and precipitation data (Daly et al., 2008). The primary value of this dataset is that it provides SWE estimates at locations other than the SNOTEL stations. This allows for the aggregation of SWE information over areas of interest (Zeng et al., 2018). The UASWE product has been shown to outperform (Dawson et al., 2018) other gridded SWE products such as the SWE estimates from the Snow Data Assimilation System (SNODAS, Center. 2004). Additionally, a spatially-continuous, gridded product allows us to build a more complete picture

of spatial changes in SwS and how changes in SwS are occurring at aggregated scales. Despite the good performance of the UASWE product, there are limitations to using any modeled SWE product. Factors such as imperfect physics, inaccurate boundary conditions and re-scaling errors can contribute to inaccuracies in the modeled field, SWE, in this case (Sturm, 2015; Zhang et al., 2007).

The EPA Level III Ecoregions (ER3s) (McMAHON et al., 2001; Omernik and Griffith, 2014), which are regions with similar ecosystems and environmental resources, were used to identify mountainous regions and to delineate the grid cells in the UASWE dataset that were associated with each ER3 (Figure 2). Mountainous ER3s were included in this study if at least half of their area resided in the snow covered mask (described in section 2.3 below). Since each ER3 has similarities in biotic, abiotic, terrestrial and aquatic ecosystem components, examining SwS change in any given ecoregion may help us understand ecosystem impacts that are related to changes in SwS. Numerous ER3s correspond to the major mountain regions in the western and eastern US that serve as the largest natural reservoirs in the country.

Finally, NASA SRTM Digital Elevation data (Farr et al., 2007) were re-gridded to create a digital elevation model (DEM) matching the grid of the UASWE product. Elevation data were used to calculate watershed hypsometry in each ER3. The procedure used to calculate a hypsometry grid is described in section 2.4.2.

Though the station and ER3 datasets extend beyond the conterminous US, the UASWE dataset does not. All datasets were spatially constrained to the conterminous US in order to facilitate the comparison of results between spacial scales. A summary of all of these datasets is provided in Table 1.

## 2.3 Study Area

As noted above, this study considers both discrete station data that focus on the western US, and spatially-continuous gridded data that cover the conterminous US. Regarding the gridded data, many locations have little to no snow. Therefore, the analysis of the gridded product is restricted to locations that have a mean of at least 30 snow covered days per year based on the 39-year climatology (1982-2021) of the UASWE dataset (Figure 3). As expected, snow cover duration increases with latitude and elevation, with the longest snow cover duration found along mountain tops in the western US. In the ER3 SwS change analysis, all ER3s are considered that contain grid cells that meet the 30-day snow cover threshold, though the mountainous ER3s are more closely examined since they store the bulk of our winter water.

## 2.4 Analysis

### 2.4.1 SwS Trends

To evaluate significant trends in $SwS_A$ and $SwS_M$ across the US, these quantities were computed over a 39-year period of study (water years 1982-2020) at stations and at UASWE grid cells, which have an area of 16 km$^2$. The grid cell-based SWE from the UASWE product was additionally aggregated for each ER3 in order to assess trends at larger scales. To compute SwS at aggregated ER3 scales, the gridded SWE data within an individual ER3 were simply integrated spatially, resulting in SwS with units of meter$^3$-days.

This study used the Hamed and Rao Modified MK test from the pyMannKendall python package to compute trends in SwS (Hussain and Mahmud, 2019). The Mann-Kendall test is a rank-based non-parametric test that is used to evaluate monotonic (increasing or decreasing) trends in temporally-varying data (Hirsch et al., 1982). Thus, the null hypothesis is that the data are randomly and independently ordered and the alternative hypothesis is that a monotonic trend exists in the data. Though the Mann-Kendall test is widely used in hydrological studies, it does not account for positive autocorrelation, which increases the

probability of detecting trends when no trends exist. Because of this, many studies have turned to a modified Mann-Kendall test that does account for autocorrelation (Hamed and Rao, 1998).

### 2.4.2    Trends by Elevation in Mountainous Ecoregions

This analysis focused on 16 ER3s corresponding to the mountain ranges that receive substantial snowfall relative to surrounding ecoregions. 12 of these ecoregions are located in the western US, and 4 ER3s are located in the eastern US. The relative

elevation of $SwS_A$ change in each ER3 is examined in this study. In order to make trends in $SwS_A$ comparable over the wide range of elevations across the US, the elevations of each ER3 are converted to hypsometry scores. Each ER3 boundary is used to select co-located elevation data from the regridded NASA SRTM Digital Elevation Dataset. ER3 hypsometry is calculated by determining the percentage of the ER3 area that falls below a given elevation within that ER3. Thus, there is 0% of the ER3 at the lowest elevation of the ER3 and 100% of the ER3 is below the highest elevation. Each elevation grid cell in the DEM is

turned into a value between 0 and 1 based on where that grid cell lies relative to other elevation grid cells within the same ER3. Hypsometry scores in each mountainous ER3 are then binned into 10% increments, from 0% of an ER3 below to 100% of an ER3 below, in order to compute the mean $SwS_A$ and the percent change in each hypsometry band from 1982-2020. The percent change in the interquartile range (IQR) of SWE was also computed for each hypsometry band from 1982-2020. To calculate the percent change in IQR, the IQR in each ER3 is calculated for each year in the study by subtracting the 25th percentile from

the 75th percentile of SWE. The trend is evaluated in each hypsometry band following the trend analysis described in section 2.4.1.

### 2.4.3    $SwS_A$ Compared to Other Snow Metrics

$SwS_A$ trends are compared to other commonly used snow metrics including April 1st SWE, $SWE_{max}$, day of $SWE_{max}$, and SCD in order to evaluate what type of information the $SwS_A$ metric provides that other metrics do not. This is done in four

ways using the station data. First, the percent of stations with positive, positive significant, negative and negative significant trends in each metric are computed. Second, the trend in the annual number of snow-free periods is calculated at each station from 1982 to 2020 to evaluate whether snowpacks are becoming more ephemeral using the Hamed and Rao Modified MK test described in section 2.4.1. The annual number of snow-free periods is defined as the number of times in a water year there is no snow following a period of snow. Next, the utility of the SwS metric compared to other snow metrics is shown using a

case study of SNOTEL station 706 (Quartz Mountain, Oregon), a station that has transitioned from a mountain snowpack to an ephemeral snowpack. Third, a regression is computed between the percent change in $SwS_A$ and each other metric above using empirical data from the stations. Finally, the relationship between the percent changes in the empirical data is compared to the

percent changes that would be expected based on the conceptual SWE curve. For example, the empirical relationship between the percent change in $SwS_A$ and the percent change in $SWE_{max}$ is compared to what it would be if there was a uniform scaling in the conceptual SWE curve as shown in Figure 1(b).

## 3  Results

### 3.1  SwS Change Trends

#### 3.1.1  $SwS_A$ Change Trends

The average $SwS_A$ across all stations in this analysis is 60 md. The lowest $SwS_A$ found at a single station was 0 md and the maximum $SwS_A$ observed was 510 md. Changes in $SwS_A$ range from a decrease of 122 md to an increase of 69 md over the period of study (Figure 4). Of the 97 SNOTEL and COOP stations with increasing trends in $SwS_A$, only 10 had significant ($p<0.1$) increases (Figure 4). Significant decreasing $SwS_A$ trends were found at 123 of the stations of the 367 stations. Losses in $SwS_A$ ranged from 2 md to 122 md. Spatially, there are widespread decreasing $SwS_A$ trends across most of the 11 western states that contain snow stations, with declines ranging from 17% to 87%. The 10 stations with significant increases in $SwS_A$ range from a 6% increase to a 78% increase. The stations with increasing $SwS_A$ trends are mostly located in the Northern and Middle Rockies and also includes a few station in the Southern Rockies and in the Cascades.

Moving from discrete station data to the the spatially-continuous gridded UASWE data, there is a mean $SwS_A$ of 1.8e8 m$^3$d across grid cells (Figure 5). Mountainous ER3s in the western US have an average $SwS_A$ of over 1.4e9 m$^3$d and the maximum $SwS_A$ is 2.6e10 m$^3$d. The average $SwS_A$ in much of New England and the Upper Peninsula of Michigan ranges from 3.2e8 to 6.4e8 m$^3$d.

The grid cell-scale change analysis yields similar geographic patterns of significant changes in $SwS_A$ in the western US as the station-scale analysis (Figure 5). This is not surprising given that the UASWE product assimilates SNOTEL (and other) station data. The benefit of including a spatially distributed product such as UASWE in this analysis is that it adds detail and insight as to where changes in $SwS_A$ are occurring beyond the western US and in-between the locations where discrete stations are located. Significant increases in grid cell $SwS_A$ are primarily found in the north-central and north-eastern US. Only 5% of US grid cells have significant increasing trends and have a mean percent increase of 84%. From 1986-2015, the north-central and north-eastern US experienced an increase in annual precipitation, particularly in spring and fall, though these regions also show spatially-variable increases in precipitation during the winter (Easterling et al., 2017). These precipitation changes may partially explain the increases in $SwS_A$, though these regions have also experienced increases in winter temperatures over the same time period. Significant decreases in $SwS_A$ are more widespread and are found across the western US, the Appalachian Mountains, the Blue Ridge Mountains and in the Ozarks. Of the 54% of US grid cells that have decreasing trends in $SwS_A$, 11% have significant decreasing trends. The mean percent decline in $SwS_A$ for the grid cells with significant trends is 44%.

Figure 6 indicates the raw change and percent change in $SwS_A$ across ER3s. Aggregating UASWE $SwS_A$ at ER3 scales spatially-filters (and thus mutes) some of the grid cell-scale trends in $SwS_A$ as can be seen when comparing Figures 4 and 5.

Of the 51 ER3s that are evaluated in this study, 19 have increasing trends and 32 have decreasing trends. Only one ER3, the non-mountainous Lake Agassiz Plain, has a significant positive trend in $SwS_A$ (86% increase). None of the non-mountainous ER3s have significant decreasing $SwS_A$ trends. The specific ways in which $SwS_A$ has changed across the 16 mountainous ER3s and how these changes relate to other snow metrics will be discussed in section 3.3.

### 3.1.2 $SwS_M$ Change Trends

The highest monthly mean $SwS_M$ occurs in March at stations (12.8 md), grid cells (2.7e7 $m^3$d) and in mountainous ER3s (7.4 $m^3$d). Figures 7, 8, and 9 summarize trends in $SwS_M$ change evaluated at stations, UASWE grid cells and ER3s, respectively. Significant decreases in $SwS_M$ occur across all months at all spatial scales examined. November experienced the highest number of stations, grid cells and ER3s with significant $SwS_M$ losses compared to any other month. The greatest monthly median percent loss of $SwS_M$ occurred in November at stations (56%) and grid cells (44%), and in March at mountainous

ER3s (61%). Looking at raw change values, the months with the greatest decrease in $SwS_M$ are not the same as the months with the largest percent changes. March, December and January are the months with largest decrease in median $SwS_M$ at stations (4.6 md), grid cells (6.4e6 $m^3$d) and mountainous ER3s (1.1e11 $m^3$d), respectively.

Though there is an overall negative median percent change in $SwS_M$ in all winter months at the station and grid-cell scale, February and March have higher occurrences of significant $SwS_M$ increase than any other months. At the grid cell-scale,

October, March-May and July-August all have a 0% median percent change in $SwS_M$ because most grid cells within the snow-cover mask are snow-free during these times. At the ER3 scale, the median percent change in $SwS_M$ is negative in all months. Most data points that indicate significant positive increases in monthly storage are considered outliers at all spatial scales.

### 3.2 $SwS_A$ Change Trends in Mountainous ER3s

Analysis of mountainous ERs illuminates the large role mountains play in storing winter snow water resources as snowpack,

particularly in the western US. An average of 72% of the annual $SwS_A$ in the US (3.5e13 $m^3$d) is held in the 16 mountainous ER3s, despite these ER3s only covering 16% of the US land area (Figure 6). Western mountainous ERs cover 12% of the US land surface and store an average of 65% of the annual $SwS_A$. Across all mountainous ER3s, there has been a 22% decline (6.1e12 $m^3$d) in $SwS_A$ over the 39 year period of study. Over the same time span, there has been a 24% decline in $SwS_A$ in western mountainous ER3s, indicating that western snow reservoirs are shrinking faster than eastern snow reservoirs.

Of the 16 mountainous ER3s (outlined in red), only the Northeastern Highlands has a (non-significant) increasing $SwS_A$ trend, while the other 15 mountainous ER3s have decreasing $SwS_A$ trends. This means that the snow water storage in 94% of mountainous ER3s has declined from 1982-2020. Five of the mountainous ER3s, the Cascades, the Eastern Cascade Slopes and Foothills, the Southern Rockies, the Idaho Batholith and the Arizona/New Mexico Mountains, have significant decreasing $SwS_A$ trends, with a mean percent decrease of 38%.

Table 2 summarizes the fraction of US $SwS_A$ in each mountainous ER3, the percent change in $SwS_A$ from water years 1982 to 2020, and the p-value associated with the percent change. In the western US, the Middle Rockies are responsible for the greatest fraction (12%) of $SwS_A$ in the country, followed by the Southern Rockies (10%) and the Idaho Batholith (8%). $SwS_A$

has declined in all mountainous ER3s in the western US over the last 39 years. The greatest declines in western SwS were in the Arizona/New Mexico Mountains (56% decline), the Eastern Cascade Slopes (40% decline) and Foothills and the Cascades (39% decline). All eastern mountainous ER3s showed declines in $SwS_A$ over the last 39 years with the exception of one, the Northeastern Highlands. The Northeastern Highlands are responsible for the greatest fraction (4%) of $SwS_A$ in the eastern US where $SwS_A$ increases 13% over the last 39 years. The greatest decline in $SwS_A$ in the eastern US was in the Ridge and Valley (11% decline), which holds an average of 0.2% of US $SwS_A$.

The greatest $SwS_A$ is found in the highest 10% of mountainous ER3s elevations (Figure 10). Most mountainous ER3s have decreasing trends in $SwS_A$ across all hypsometry bins, though the Sierra Nevada (5), the Wasatch and Uinta Mountains (19), the Southern Rockies (21), the North Central Appalachians (62) and the North Cascades (77) show increasing trends in $SwS_A$ at low elevations with decreasing trends at higher elevations. The Northern Highlands is the only ER3 that shows increasing trends in $SwS_A$ at all elevations. Increasing trends in $SwS_A$ at low elevations in some ER3s may partially be a result of very low $SwS_A$ to begin with, thus small changes in $SwS_A$ may suggest large percent changes.

By looking at the percent change trends in the IQR of SWE, this study gives an idea of how interannual SWE variability has changed from 1982-2020 (Figure 10). Several ER3s have increases in the SWE IQR of the lowest hypsometry bands, which correspond to the lowest parts of ER3s. This could be a result of increasing snow variability as freezing levels move to higher elevations, resulting in increased irregularity in precipitation form. In the middle and upper hypsometry bands of most ER3s, there is largely a decrease in the IQR. This may be a result of declining snowpacks, which would allow for less variability in the range of SWE values overall. The Northern Highlands, Ridge and Valley, Central Appalachians and North Cascades stand out in that they have increasing tends in IQRs across most hypsometry bands.

### 3.3   $SwS_A$ Compared to Other Snow Metrics

Figure 11 demonstrates how the conceptual SWE curve is changing in each of the ER3s based on the trend analysis of four common snow metrics; DSO, $SWE_{max}$, $SWE_{max}$ day of water year ($D_{max}$), and DSD. Trends in DSO, $SWE_{max}$, $D_{max}$, and DSD were evaluated because these metrics serve as anchor points that define the boundaries of the conceptual SWE curve. These conceptual SWE curves are superimposed on the observed mean SWE curves for the first and last 20 years of study in each ER3. With the exception of the Northern Highlands and the North Cascades, the 2020 SWE curve (solid red line) delineates a smaller conceptual SWE curve than in 1982 (dotted red line). The specific anchor points that cause the shrinkage of the conceptual SWE curve are variable across ER3s. Of the ER3s that experienced significant ($p < 0.1$) decreases in $SwS_A$, the Southern Rockies and the Arizona/New Mexico mountains also had significant changes in the DSO, DSD and $SWE_{max}$. The Cascades, the Eastern Cascade Slopes and Foothills and the Idaho Batholith also experienced significant declines in $SwS_A$. These three ER3s had significantly earlier DSDs. Although evaluating change at the anchor points tells us how the conceptual SWE curve is changing, the actual SWE curve is not a triangle and is subject to complex patterns of change including notable accumulation- and ablation-season SWE variability. A comparison of the mean SWE curves from beginning and end of the study period yields a smaller SWE curve in the last decade of study in all western ER3s, though the eastern ER3s (Northeastern Highlands, North Central Appalachians, Ridge and Valley and Central Appalachians) have more nuanced change in the SWE

curve. These subtler shifts involve certain aspects of the snow season having higher SWE values in the first decade of study while other parts of the snow season have higher SWE values in the later period of study. The SwS metric accounts for these complex patterns and is able to quantify natural reservoir storage and change.

Trends in snow metrics ($SwS_A$, April 1st SWE, $SWE_{max}$ and SCD) are not changing in the same direction at all stations (Figure 12). SNOTEL and COOP stations were placed to capture mountain snowpack regimes where snow increases up to a maximum value throughout the accumulation season and then disappears across the ablation season. April 1st SWE, $SWE_{max}$, SwS and SCD are snow metrics that are useful for characterizing and monitoring change in mountain snowpack regimes and describe points relevant to a conceptual SWE curve. While trends in these metrics have changed in the same direction at 286

of the 465 SNOTEL and COOP stations, 38.5% of stations have non-uniform inter-metric trend directions. This means that if trends in only one snow metric were to be examined, it could paint an incomplete picture of change.

The inability of common one-dimensional snow metrics to reflect snow storage change is particularly apparent when snowpacks transition from one snow regime to another, such as a permanent snowpack transitioning to a mountain snowpack or a mountain snowpack transitioning to an ephemeral snowpack. In our observational record, this study finds that snow regimes

are becoming increasingly ephemeral at many station locations. This study found the number of annual snow-free periods has significantly increased at 23% of stations over the last 38 years. The Quartz Mountain (OR) SNOTEL station (706) provides an example of where a snowpack has increased in ephemerality (Figure 13). This station went from an average of 1.7 snow-free periods per water year over the first decade of study to an average of 6.3 snow-free periods per water year over the last decade of study. A side-by-side comparison of trends in $SwS_A$, April 1st SWE, $SWE_{max}$ and SCD, illustrates how the SwS metric is

able to capture complex change, such as is associated with an ephemeral snowpack. At the Quartz Mountain station, there are significant negative trends in SwS and $SWE_{max}$, no trend in April 1st SWE and a positive trend in SCD (Figure 13). In this example, April 1st SWE is largely not relevant as a snow monitoring metric because the majority of years are snow-free on April 1st at this location. This station is interesting because it has opposite significant trends in $SWE_{max}$, which is decreasing, and SCDs, which are increasing. Since $SwS_A$ is the integral of the SWE curve, both magnitude and duration of snow cover are

incorporated into its calculation. This allows the SwS metric to provide a robust picture of the degree to which the snowpack is serving as a reservoir for water storage and how that reservoir function may be changing.

There are other ways to demonstrate the utility of SwS as an additional tool for gaining insight into our changing snowpack. April 1st SWE is overwhelmingly the metric cited by water resource managers as the singular measure of the season's snow. But, how do changes in that measure correspond to changes in others? First, Figure 14 shows the relationships between percent

changes in various snow metrics of interest. It can next be useful to return to the idea of a conceptual SWE curve (Figure 1(a)). Based on this simple geometry, the $SwS_A$ is given by

$$SwS_A = \frac{1}{2} SWE_{max} SCD. \tag{2}$$

If the geometry of the conceptual SWE curve were to be preserved over time, the SWE curve would be uniformly scaled (Figure 1(b)). In this scenario, one would expect the percent change in $SWE_{max}$ and SCD to each be half the percent change

in SwS. However, the regression plots in Figure 14 reveal that the percent change in SCD is roughly 26% that of the percent

change in SwS. And it is found that the percent change in $SWE_{max}$ is 86% that of the percent change in $SwS_A$. What this means is that the conceptual SWE curve has been flattening over the period of study of the data at stations. So, relying on a single metric like April 1st SWE gives an incomplete assessment of the storage of snow throughout a full season, and a more holistic metric like SwS may be more informative when considering a full snow season.

## 4 Discussion

SwS is a unique snow metric because it essentially has unlimited degrees of freedom - any change in the SWE curve (e.g. changes in SWE magnitude, timing, variability, etc.) will be captured in its calculation. Thus, regardless of how the SWE curve changes, the SwS metric is able to provide information about water storage in natural reservoirs at a given location. SwS is different from the other common snow metrics discussed in this paper (April 1 SWE, SCD, $SWE_{max}$) because the other metrics have 1 degree of freedom - they provide data on one dimension of the SWE curve. The flexibility of the SwS metric is particularly useful when attempting to quantify storage in snowpacks that have fundamental shifts in their SWE curve. As demonstrated above, the SwS metric can still quantify natural reservoir storage when a snowpack has transitioned from a mountain-type snowpack to an ephemeral type. If a snowpack were to transition from a permanent snowpack to a mountain-type snowpack, the SwS metric would also be able to provide information on the storage.

Of the snow metrics discussed in this paper, SwS is uniquely positioned to capture storage change at aggregated scales, across the full SWE curve. Metrics such as DSO, $SWE_{max}$, DSD, SCD, etc. are essentially anchor points for the conceptual SWE curve. Though it can be beneficial to note changes at any of these points, change can also happen in-between these anchor points, such as an increase in accumulation season ablation or an increase in ablation season accumulation as is seen across mountainous ER3s in Figure 11. This type of change has already been documented as changing melt fractions (Musselman et al., 2021). At aggregated scales, such as across a watershed, snowpack variability (due to landscape features such as aspect or elevation) influences the SWE curve between the anchor-point metrics. Change that occurs between the anchor-point metrics is not inherently captured by these metrics. Since the SwS metric can accommodate various spatial scales, it is able to capture natural reservoir storage regardless of variability in snow change or snowpack fluctuations that occur in-between anchor point metrics.

The widespread losses of $SwS_A$ over the last 39 years reported in this study are consistent with the broader narrative of snowpack change literature, which has established declines in snow covered area, snow cover duration, April 1st SWE, $SWE_{max}$, etc (Rupp et al., 2013; Mote et al., 2018; Notarnicola, 2020; Marshall et al., 2019; Bormann et al., 2018; Choi et al., 2010; Huning and AghaKouchak, 2018). Losses of winter snowpack are largely attributed to increasing global temperatures (Hamlet et al., 2005), which have resulted from a combination of natural variability and anthropogenic-caused climate warming (Rupp et al., 2013; Pederson et al., 2013). The declining trends in $SwS_A$ are a reflection of declining trends in $SwS_M$ in nearly every month, at every scale. The greatest percent losses in $SwS_M$ occur early in the snow season, particularly in November. The loss of early season SwS is consistent with previous work that used satellite imagery and reported that DSO is occurring later (Notarnicola, 2020). While future work could explore the exact mechanistic drivers of predominantly decreasing $SwS_A$ trends,

these findings are reasonable in the context of mechanistic drivers explored in other snow change literature. From an energy budget standpoint, snow falling at warmer temperatures (as a result of climate warming) and overall shallower snowpacks (due to reduced snowfall fractions) contribute to reduced cold content and more readily ripening snowpacks (Jennings and Molotch, 2020). Additionally, shallower snowpacks are susceptible to enhanced snowmelt from the albedo feedback as vegetation and soil are exposed (Kapnick and Hall, 2010). Though the majority of $SwS_A$ trends are declining, the north central plains and the Northeastern Highlands show increasing $SwS_A$ trends. Snowmelt and rain-on-snow are known to be flood generating mechanisms in New England, Minnesota and along the Mississippi and Missouri Rivers (Collins, 2009; Novotny and Stefan, 2007; Wiel et al., 2018; Olsen et al., 1999). The increasing SwS trends in these regions may therefore have implications for flood hazards.

Spatial scale has long been a topic of conversation in snow hydrology as certain processes that occur at very small scales contribute to considerable within-grid cell heterogeneity as one scales up from point to grid cell to regional scales (Blöschl and Sivapalan, 1995; Molotch and Bales, 2005). This works finds differences in the magnitude and timing of significant changes in $SwS_A$ and $SwS_M$ when different spatial scales are compared. For example, less of the US landscape shows significant changes in $SwS_A$ in the ER3 analysis compared to the grid cell analysis. Thus, the aggregation of $SwS_A$ into ER3s filters some of the grid cell-scale spatial $SwS_A$ trends. Temporally, there is a higher fraction of sites with a significant positive increase in $SwS_M$ from October - March in the grid cell analysis compared to the ER3 analysis. This indicates that local significant increases in $SwS_M$ at grid cell-scales are offset by smaller magnitude increases in $SwS_M$ or decreases in $SwS_M$ at many locations once the $SwS_M$ is aggregated to ER3 scales. From a water resources perspective, these findings underscore the importance of choosing an appropriate aggregation scale in order to accomplish management goals.

In the western US, where snowmelt is vital to supplementing warm-season water supplies, about 70% of runoff in mountainous regions originates as snow (Li et al., 2017). Snowpacks also play an important role in climate, ecological processes and recreation in both eastern and western mountains. Across all mountainous ER3s, there has been a 22% decline in $SwS_A$ over the 39 year period of study. The ER3 mountain ranges considered in this work include the headwaters to 13 of the 18 water basins located in the US, underscoring the importance of these natural reservoirs to water resources. The loss of SwS in these regions is of further concern as the warm season is projected to increase in length due to anthropogenic climate warming (Mallakpour et al., 2018; Padrón et al., 2020; Siler et al., 2019). Siler et al. (2019) also suggests that declining snow trends may accelerate once the current natural climate mode changes because natural variability has slowed the decline of western snowpacks since the 1980s. The capacity of natural snow reservoirs is declining in most of the western US and across most mountain ranges in the US. Since declining trends are expected to continue into the future, monitoring our natural snow reservoirs is essential. Metrics that are highly flexible in space and time (like SwS) can be used in monitoring change and evaluating future projections.

$SwS_A$, the changes in $SwS_A$, and the variability in SWE are all influenced by elevation. The greatest amount of snow water storage occurs on a disproportionately small fraction of our landscape - at the highest elevations of mountainous ER3s. Almost all mountainous ER3s are losing $SwS_A$ at all elevations. In the majority of mountainous ER3s, the highest elevations have experienced the greatest losses $SwS_A$ over the last 39 years. The elevation-dependent changes in our natural snow reservoirs

are likely associated to documented elevation-dependant changes in temperature and precipitation (Wang et al., 2014; Harpold et al., 2012; Pepin et al., 2015, 2022; Qixiang et al., 2018). Winter temperatures have increased significantly in the recent past (Vose et al., 2017), which increases the vapor pressure deficit in the atmosphere and may enhance sublimation and vapor fluxes (Harpold et al., 2012). Higher elevations have also warmed at faster rates than their low elevation counterparts, where there have been increasing trends in precipitation (Wang et al., 2014; Pepin et al., 2022). Wang et al. (2014) suggests that elevational warming amplification is likely associated with effective moisture convection. These mechanistic drivers are a plausible explanation for finding the greatest $SwS_A$ loss at the highest elevations.

This work also finds elevation-dependant changes in SWE variability. SWE variability has likely increased as a result of winter freezing levels moving to higher elevations (Catalano et al., 2019), an increased fraction of precipitation falling as rain instead of snow and more rain falling on snow (McCabe et al., 2007), all of which are related to increasing winter temperatures. Decreases in SWE variability at higher elevations, where there are declining trends in $SwS_A$, may be a result of shallower snowpacks overall.

## 5 Conclusions

A new snow metric, SwS, is defined and used to identify where and to what extent water storage in natural snow reservoirs has already changed in the observational record. Mountains, especially western mountains, play an disproportionate role in natural water storage relative to the surrounding landscape. High-elevation natural snow reservoirs are responsible for the greatest $SwS_A$, and have generally experienced the greatest declines in $SwS_A$. Declines in $SwS_A$ are associated with a fundamental shift in the shape of the conceptual SWE curve as it appears to be flattening across stations. As we move into a future of increased snow variability, diminished snowpacks and as more of the winter snow landscape transitions to ephemeral regimes, temporally static metrics such as April 1 SWE and $SWE_{max}$ may become less representative of our snowpacks. Concurrently, it may be useful to have metrics such as SwS that can adapt to a wide range of circumstances. Spatially and temporally flexible metrics such as SwS may become increasingly valuable particularly when it comes to monitoring change.

Declining storage in our natural snow reservoirs has broad implications for human and ecological systems. Natural snow reservoirs help to increase water storage far beyond the capacity of man-made reservoirs in the western US, supporting their roll in linking cool-season precipitation to warm-season water demand. As one of the most robust projected impacts of climate change is a continued increase in air temperatures, it is likely that declining trends in $SwS_A$ will continue. Water managers, planners and decision makers will need to account for these declines in natural snow water storage as they relate to streamflows for fish migration and recreation, municipal and agricultural water supplies and flood hazards. Though this paper does not focus on future predictions of snowpack, SwS could be a useful tool for understanding how our natural snow reservoirs change in the future. Change in our natural snow reservoirs is multidimentional and already happening. Metrics are needed that can capture this complexity of change.

*Data availability.* Below is an enumerated list providing links to the publicly available datasets used in this study.

1. SnoTel (https://wcc.sc.egov.usda.gov/reportGenerator/)

2. UASWE (https://nsidc.org/data/nsidc-0719/versions/1)

3. NASA SRTM (https://lpdaac.usgs.gov/products/srtmgl1v003/)

4. USGS WBD (https://datagateway.nrcs.usda.gov)

5. EPA Ecoregions (https://www.epa.gov/eco-research/level-iii-and-iv-ecoregions-continental-united-states)

*Author contributions.* CMA and DFH designed the research questions. CMA determined the methodology and conducted the analysis. DFH conceptualized the SwS metric. CMA prepared the paper, with guidance and feedback from DFH.

*Competing interests.* The authors declare that they have no competing interests.

*Acknowledgements.* This work was made supported by funding from the Graduate School Oregon Lottery Award for Academic Excellence from Oregon State University and the Alumni Award from the Water Resources Graduate Program at Oregon State University. We thank P.C. Loikith and L.R. Hawkins for their contributions to this research.

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

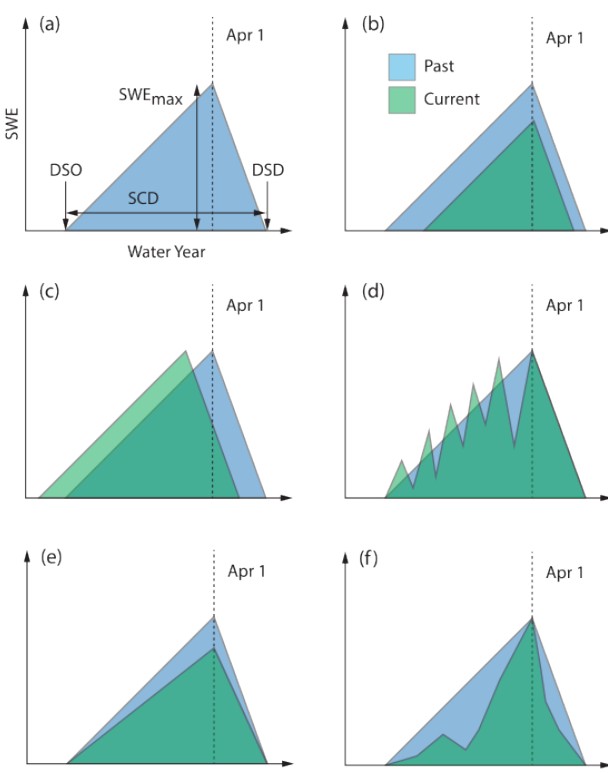

**Figure 1.** Conceptual illustration of a SWE curve and various ways that it could change from past to present.

**Table 1.** Summary of data used in this work.

| Dataset | Hosting agency | Data Type | Temporal resolution | Spatial resolution |
|---------|----------------|-----------|---------------------|--------------------|
| Snow Telemetry (SnoTel) | Natural Resources Conservation Service (NRCS) | Point observations | Stations were selected with a period of record starting in water year 1982 and less than 10 percent of days missing | N/A |
| Cooperator Snow Sensors (COOP) | Natural Resources Conservation Service (NRCS) | Point observations | Stations were selected with a period of record starting in water year 1982 and less than 10 percent of days missing | N/A |
| University of Arizona SWE (UASWE) | National Snow and Ice Data Center (NSIDC) | Gridded product | water year 1982-present | 4km x 4km |
| NASA SRTM Digital Elevation | Google Earth Engine (GEE) | Gridded product | N/A | 30m x 30m |
| EPA Level III Ecoregions | Google Earth Engine (GEE) | Vector data | N/A | N/A |

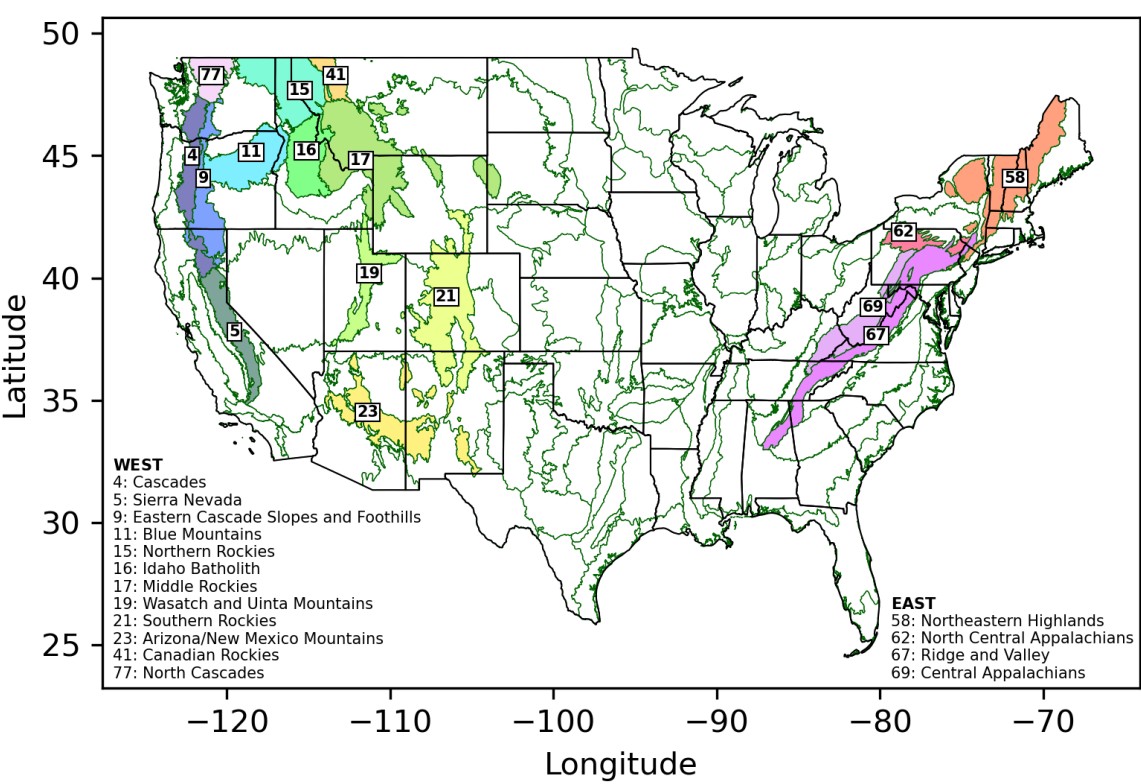

**Figure 2.** Map of ER3s in the US. Mountainous ER3s are colored and labeled.

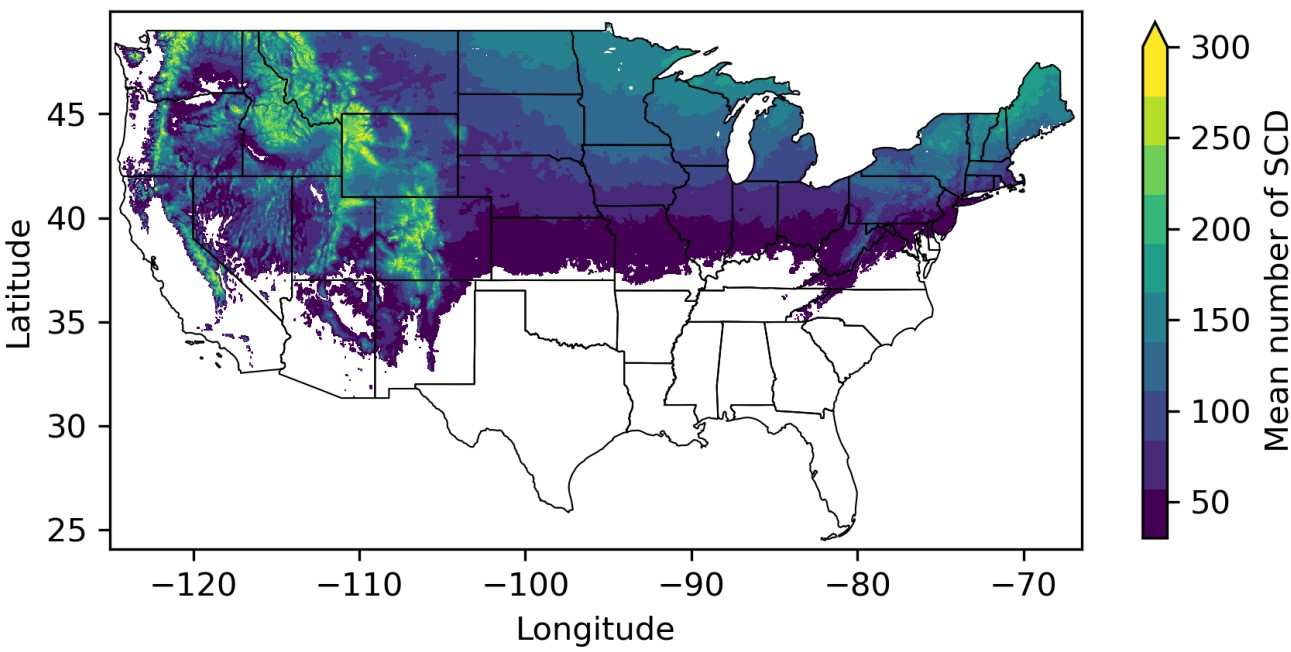

**Figure 3.** Study area for the UASWE dataset indicated by color shading showing the mean number of SCDs across the contiguous US in locations that have a minimum average of 30 snow covered days/year over the period of study.

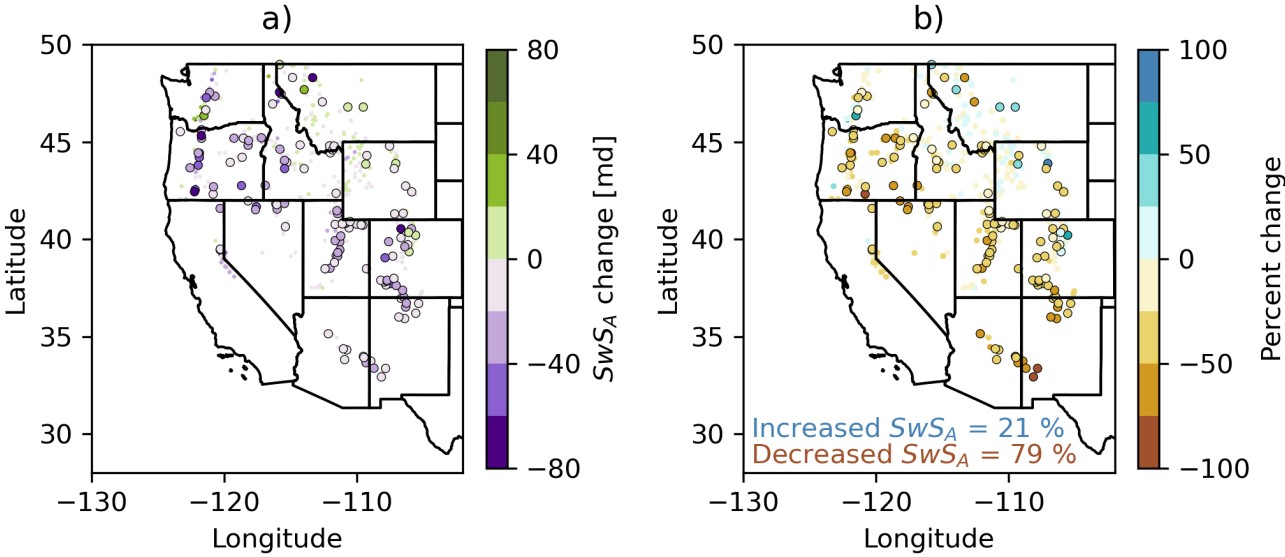

**Figure 4.** Change in SwS$_A$ [md] (a) and percent change in SwS$_A$ (b) across US stations from water years 1982-2020. Large outlined circles indicate stations with $p < 0.1$.

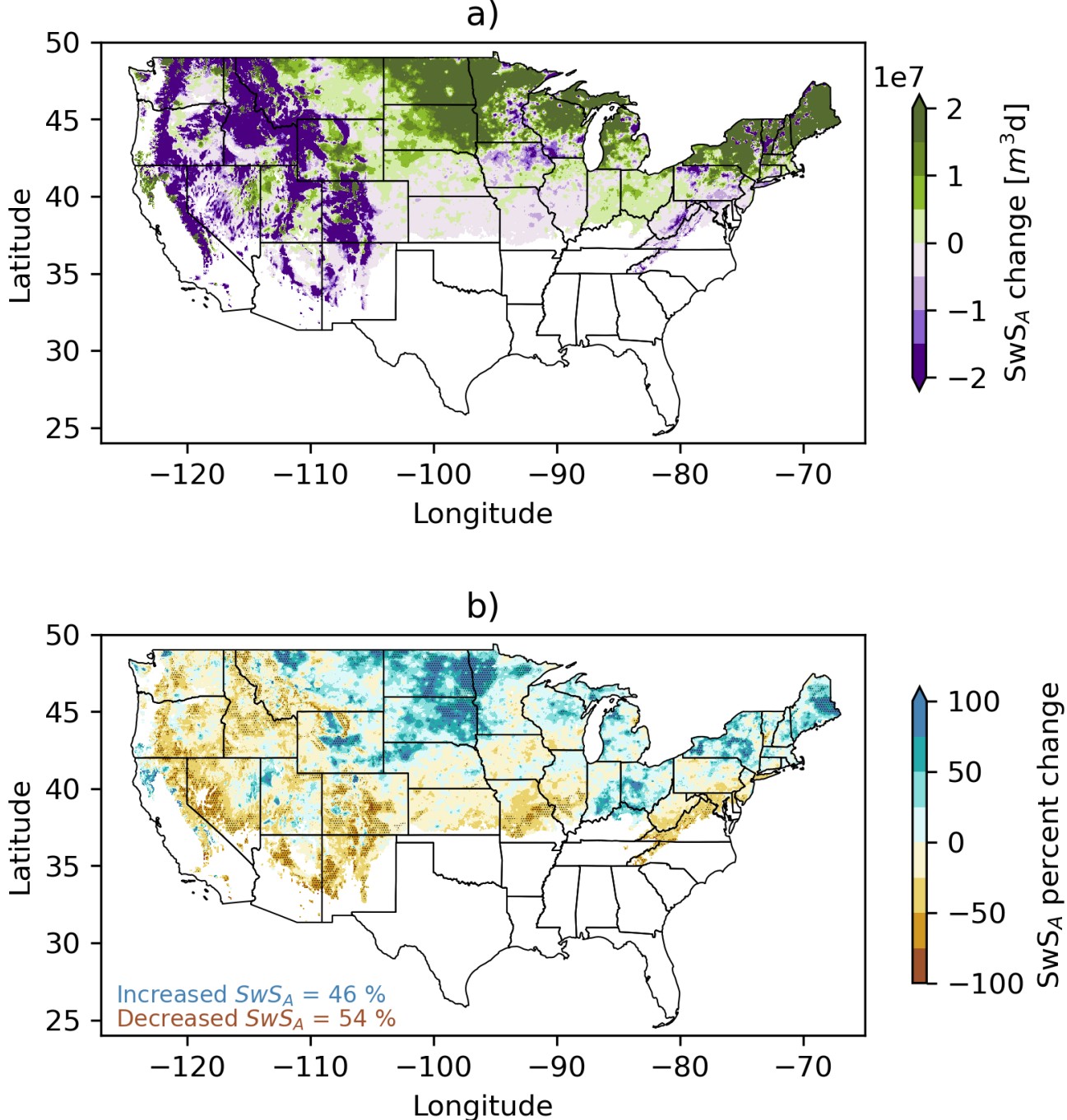

**Figure 5.** Change in SwS$_A$ [md] (a) and percent change in SwS$_A$ (b) across the UASWE dataset. Stippling indicates locations with $p < 0.1$.

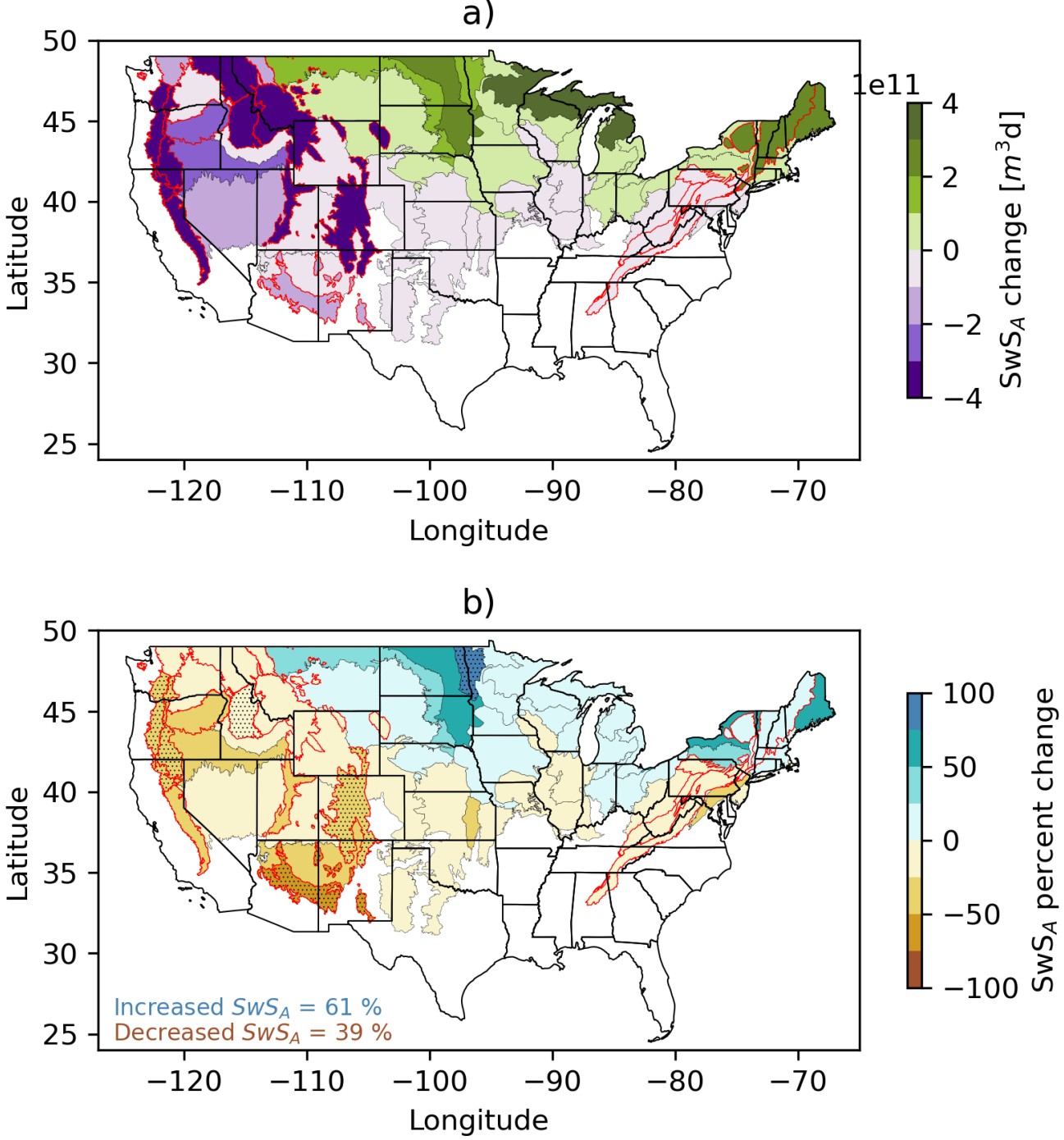

**Figure 6.** Change in SwS$_A$ [md] (a) and percent change in SwS$_A$ (b) aggregated across ER3s over water years 1982-2020. Stippling indicates ER3s with $p < 0.1$.

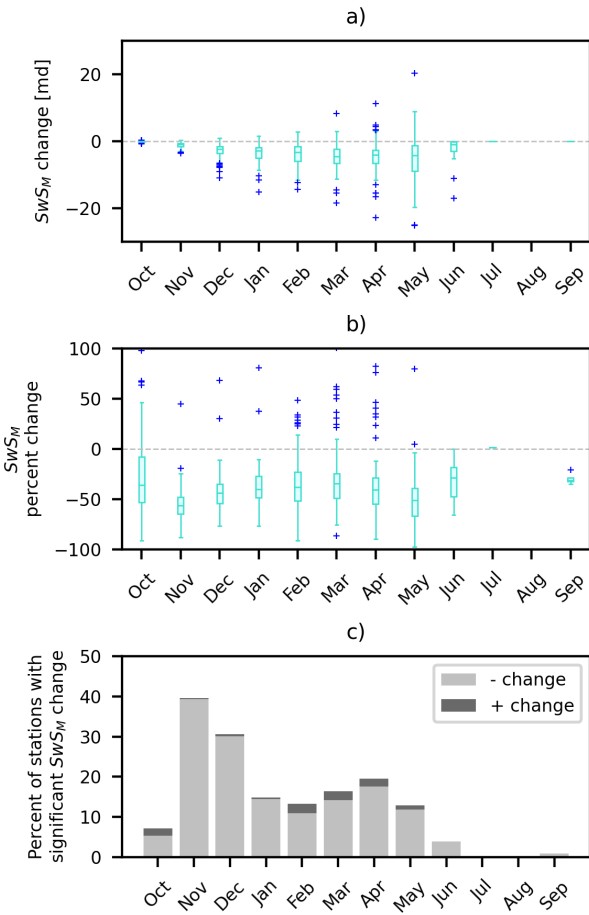

**Figure 7.** Changes (significance $p < 0.1$) in SwS$_M$ across US stations, in dimensional (a) units and in terms of percent change (b). The rectangle indicates the interquartile range, with the middle bar indicating the median. The blue pluses are outlier points. Panel (c) shows the fraction of stations that had significant increases or decreases in SwS.

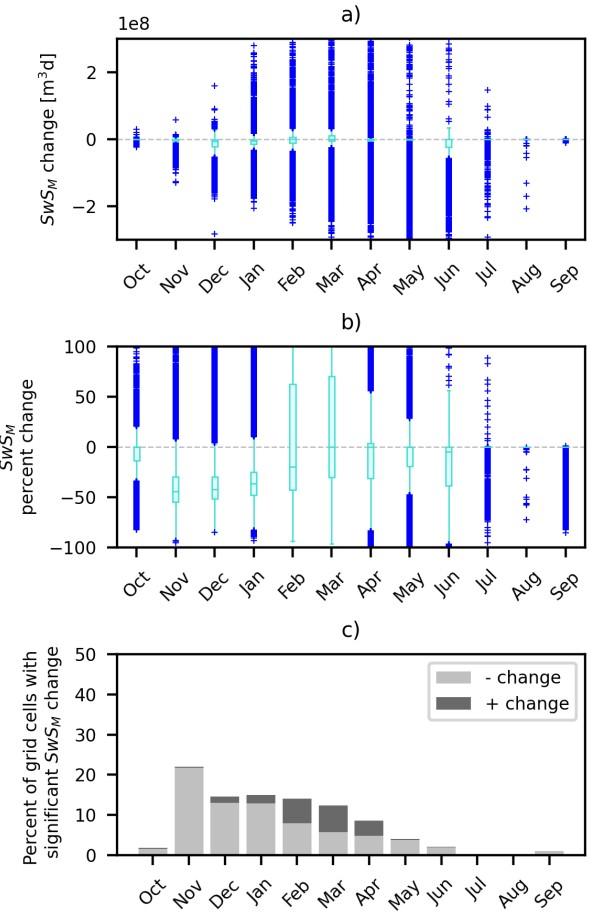

**Figure 8.** As in Figure 7, but for the gridded UASWE dataset.

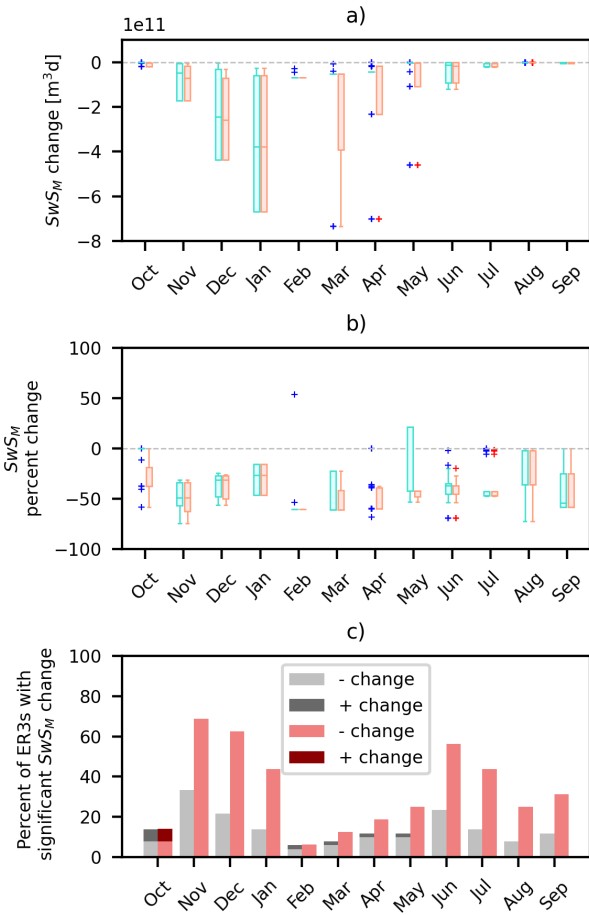

**Figure 9.** As in Figure 7, but for changes in SwS$_M$ aggregated across ER3s from water years 1982-2020. Red components of this Figure indicate the results only considering mountainous ER3s.

**Table 2.** Overview of SwS$_A$ in mountainous ER3s. Bold p-values are significant at $p < 0.1$.

| ER3 code and name | ER3 elevation [m] (minimum, maximum) | Average fraction of US SwS$_A$ | Percent change from 1982 to 2020 | p-value |
|---|---|---|---|---|
| *West* | | | | |
| 4: Cascades | 4, 3531 | 0.078 | -39.0 | **0.08** |
| 5: Sierra Nevada | 402, 3910 | 0.036 | -37.7 | 0.23 |
| 9: Eastern Cascade Slopes and Foothills | 73, 2622 | 0.023 | -38.6 | **0.09** |
| 11: Blue Mountains | 386, 2647 | 0.034 | -25.6 | 0.16 |
| 15: Northern Rockies | 368, 2106 | 0.08 | -24.8 | 0.15 |
| 16: Idaho Batholith | 618, 3298 | 0.083 | -22.3 | **0.09** |
| 17: Middle Rockies | 937, 3796 | 0.116 | -18.7 | 0.19 |
| 19: Wasatch and Uinta Mountains | 1107, 3671 | 0.032 | -29.1 | 0.14 |
| 21: Southern Rockies | 1473, 4032 | 0.098 | -33.5 | **0.001** |
| 23: Arizona/New Mexico Mountains | 805, 3451 | 0.009 | -56.2 | **0.02** |
| 41: Canadian Rockies | 973, 2550 | 0.039 | -12.7 | 0.33 |
| 77: North Cascades | 48, 2359 | 0.072 | -6.7 | 0.66 |
| *East* | | | | |
| 58: Northeastern Highlands | 23, 1436 | 0.035 | 13.1 | 0.68 |
| 62: North Central Appalachians | 170, 755 | 0.002 | -2.4 | 0.94 |
| 67: Ridge and Valley | 61, 1315 | 0.002 | -11.5 | 0.73 |
| 69: Central Appalachians | 215, 1337 | 0.003 | -9.8 | 0.68 |

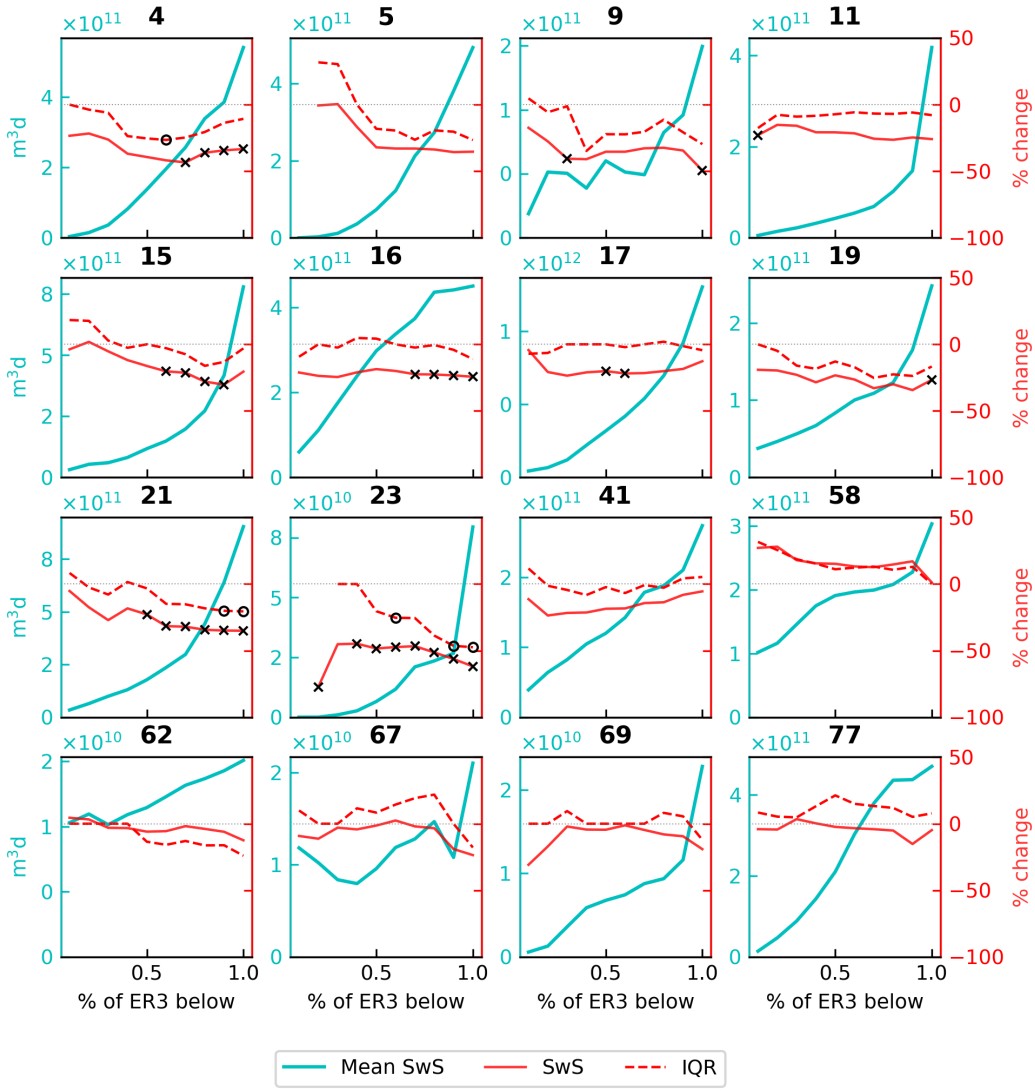

**Figure 10.** SwS$_A$ in each mountainous ER3 as a function of ER3 hypsometry. Teal line indicates the average SwS$_A$ in each hypsometry bin from water years 1982-2020 (left axis). Solid red line indicates the percent change in $_A$ as a function of hypsometry for each mountainous ER3 over the time period of study (right axis). Black × symbols indicate where the percent change in SwS$_A$ is significant ($p < 0.1$). Dashed red line indicates the percent change in the IQR of daily SWE as a function of hypsometry in each mountainous ER3 over the time period of study (right axis). Black o symbols indicate where the percent change in the IQR is significant (p<0.1). Refer to Table 2 for ER3 names.

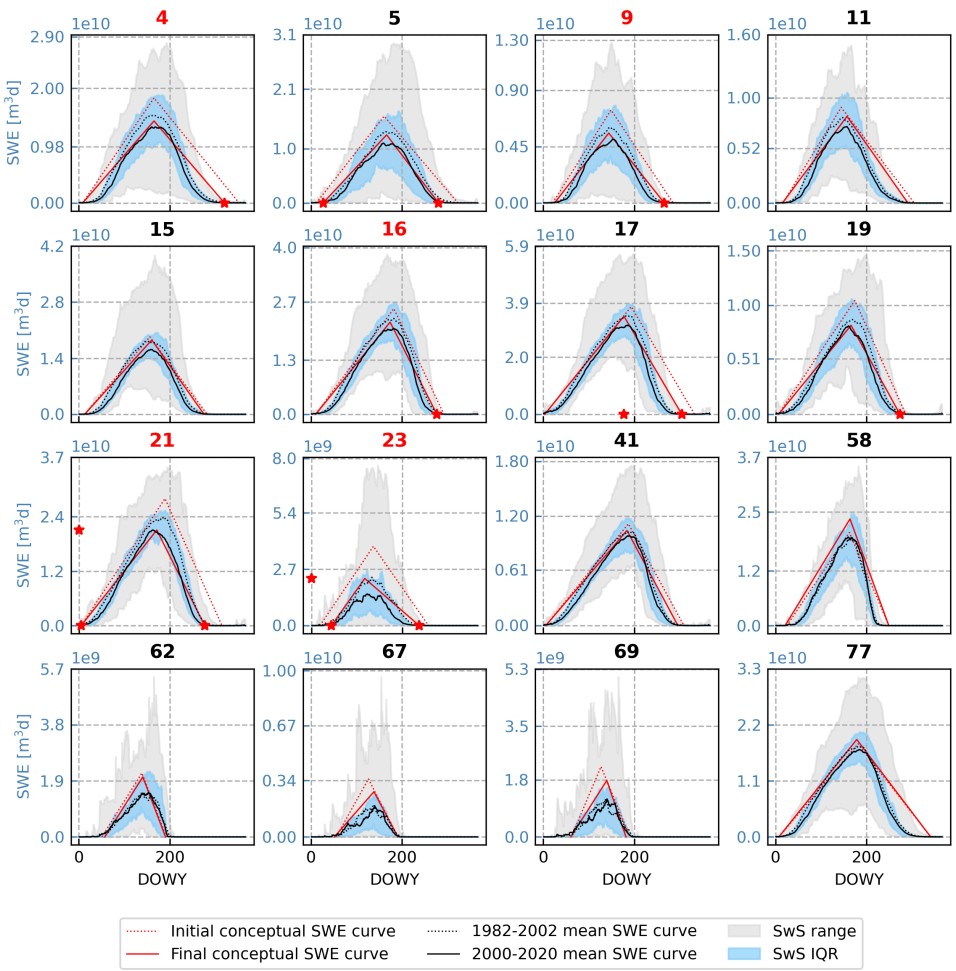

**Figure 11.** Observed SWE curves for mountainous ER3s shown with historic and current conceptual SWE curves. The black dotted line indicates the mean SWE curve from the first 20 years of our study period in each ER3 (1982-2002). The solid black line indicates the mean SWE curve from the last 20 years of our study period for each ER3 (2000-2020). Light blue shading indicates the interquartile range of observed daily SWE and grey shading indicates the minimum and maximum range of observed daily SWE. Red ER3 labels signify ER3s that had a significant decrease in SwS$_A$ over the period of study. The red dotted line indicates the conceptual SWE curve for each ER3 at the start of our study period (1982) based on the trend analysis of DSO, SWE$_{max}$, D$_{max}$, and DSD. The solid red line indicates the conceptual SWE curve for each ER3 at the end of our study period (2020). Red stars on the x-axis indicate significant decreases in DSO, D$_{max}$, and DSD. Red stars on the y-axis indicate significant decreases in SWE$_{max}$. There were no significant increases in any of the snow metrics. Refer to Table 2 for ER3 names.

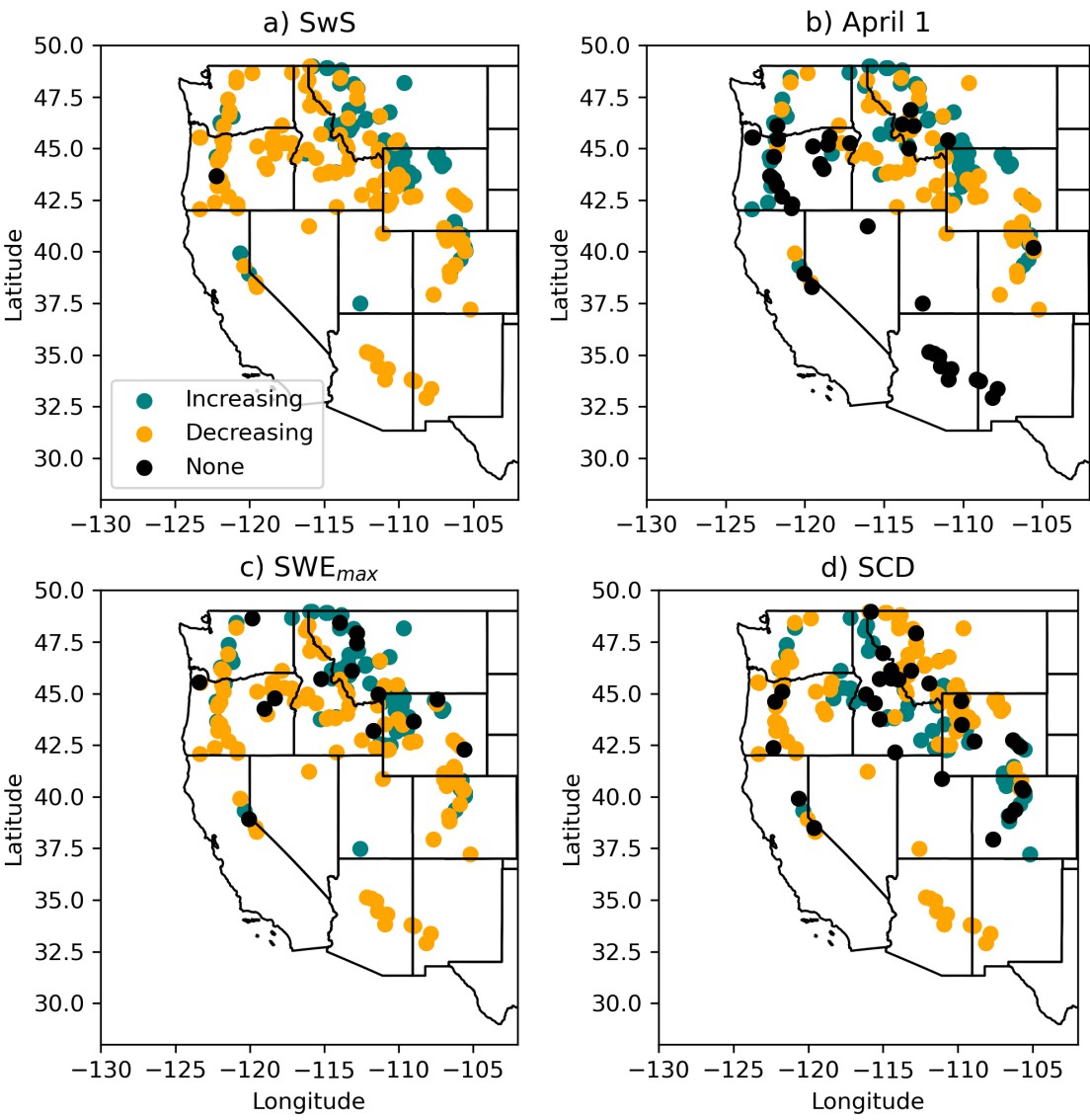

**Figure 12.** Trend directions in SwS$_A$ (top left), April 1st SWE (top right), SWE$_{max}$ (bottom left) and SCD (bottom right) across stations that do not have the same trend directions across all metrics.

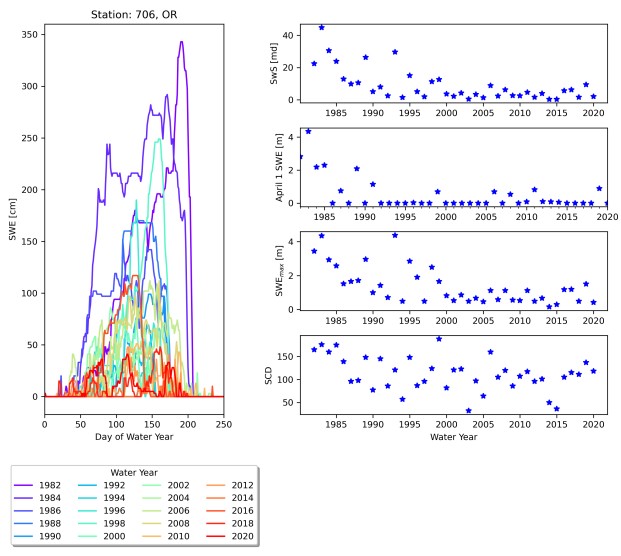

**Figure 13.** SWE curves from every second water year from 1982-2020 at the Quartz Mountain SNOTEL station.

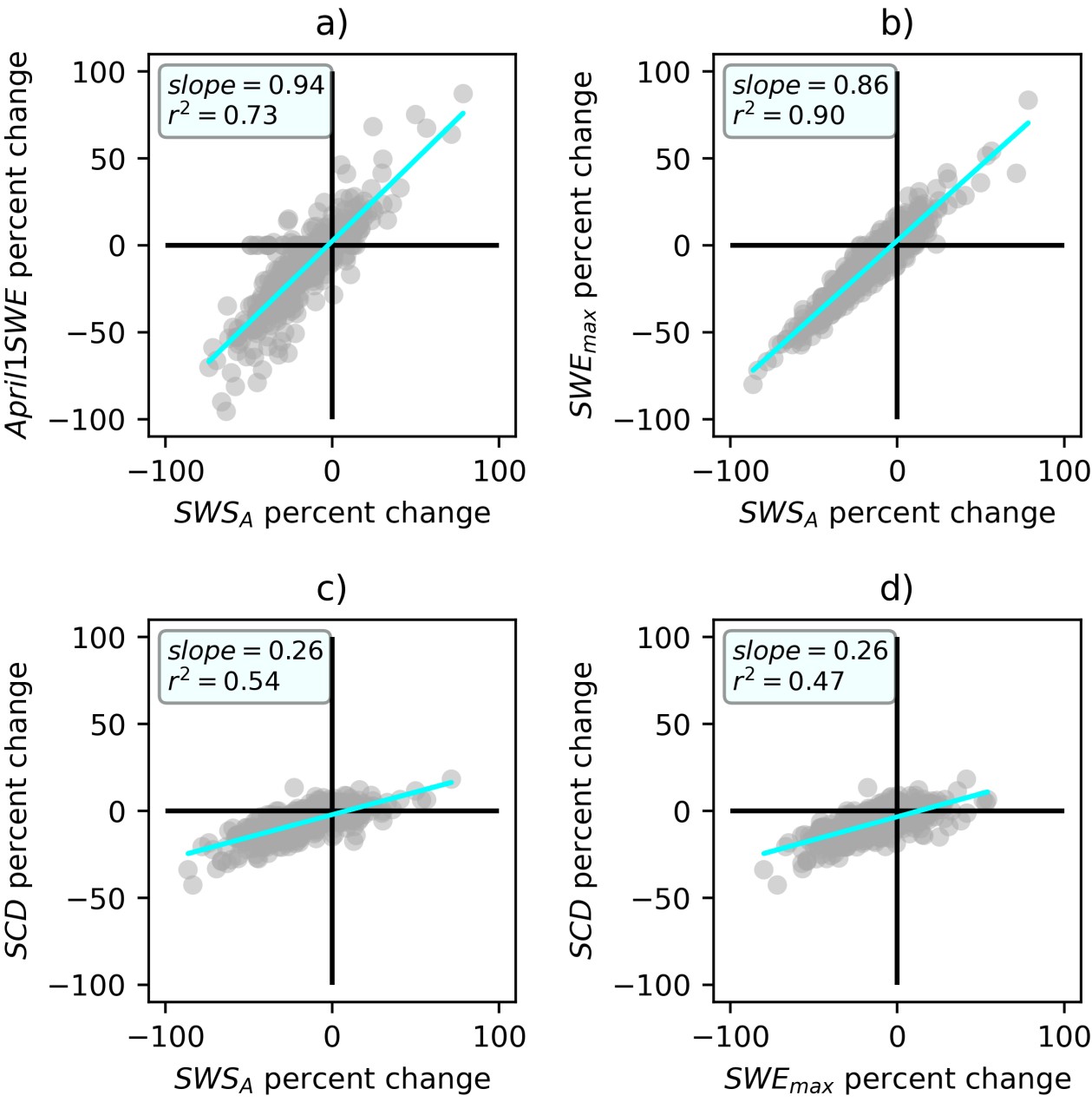

**Figure 14.** Regression of percent change in SwS$_A$ with percent change in April 1 SWE (a), SWE$_{max}$ (b) and SCD (c). Regression of SWE$_{max}$ with SCD (d).