# Peer review of "Changing Snow Water Storage in Natural Snow Reservoirs"

_EGUsphere, 2023_

## Referee Comment (RC1)

**Review: Changing Snow Water Storage in Natural Snow Reservoirs**

**Summary**
The authors of "Changing Snow Water Storage in Natural Snow Reservoirs" (Aragon and Hill) present a new metric, Snow Water Storage (SwS) to evaluate the snowpack in the mountainous United States (U.S.) throughout the length of the snow season in meter-days. Unlike traditional metrics used to quantify and characterize the snowpack at a single point in time (e.g., April 1 or peak SWE), the SwS captures the area under the SWE curve to illustrate differences and changes in snowpack accumulation and ablation seasons as well, better illustrating the nature of the complete snow season (in a given area). While the metric has great and complementary utility in quantifying the snowpack (on a monthly, annual, or by elevation bin scale), the manuscript would benefit from further depicting the SwS using actual examples across the U.S. in raw units (meter-days as opposed to predominantly reported % changes). Hypothetical examples of the SwS and changes in the SwS are presented in Figure 1 of the manuscript, yet observed and modeled changes in SwS are reported as only % changes. In order to contextualize these changes and further emphasize the added utility of this metric, the readership needs to learn how the SWE curve has changed in various parts of the U.S. to understand why the SwS has increased or decreased, and how the SwS thus provides more/added information compared to other metrics. Translating what is presented in Figure 1 to the real/raw observed and modeled data which are used in the presented work is a critical missing component to this work and would add more intuition around the new metric. Toward the tail end of the discussion, readers learn that "the conceptual SWE curve has been flattening over the 39-year period of record," which is the first mention of how the SWE curve has changed (not just monthly or annual SwS % changes), by way of the SwS evaluation, and thus provides valuable, new information (but also leaves the reader questioning, for example, how is this different from a lower April 1 SWE value? What more does this tell us about the changing snow season?). These questions need to be directly addressed (and seemingly can be, by way of the information gained from SwS). Since the metric leans on the important of temporal changes in the snow season – in addition to changes in magnitude, and thus a novel combination of snowpack characteristics – the changes in SWE curve shape need to be reported throughout the manuscript when % changes are stated (with complementary figures, ideally). This will greatly assist the readership reach the intended conclusions made in the manuscript. The manuscript would benefit from further elaborating on other, recent metrics aimed at quantifying snow water storage (e.g., Hale et al., 2023; Immerzeel et al., 2020) and being more specific in naming changes seen within individual ER3s (instead of "only one" or "four" ER3s, the authors should state the specific areas of reference), such that comparisons can be further made between the SwS and observed changes in other metrics.

**Line-by-line**
Line 7: "An average of 72% of the annual $SwS_A$" – This is a challenging number to capture as a reader. Perhaps defining/clarifying the SwS in terms of units (in the abstract) would help.

Line 20: Suggest eliminating "at river headwaters," since snow functions as a reservoir to some degree anywhere in a watershed/landscape.

Line 10: "The greatest SwSM loss occurs early in the snow snow season, particularly in November" – suggest stating the % loss or magnitude of loss, similarly with the next statement regarding "more spatially widespread significant decreases… than increases" – adding numerical statistics (as either a percent or raw value) would help contextualize the results here.
   - Also, delete "snow" where used twice in a row
   - Was the p-value for determining significance 0.05? If so (or if not), please clarify here.

Line 19: "Keystone," since there are many definitions to this term (including political and ecological perspectives, both of which the authors speak to later in this paragraph), suggest using a more precise term that is more synonymous with what is intended – or further defining this term.

Line 30: Suggest elaborating on why snow water storage may be changing, further shedding light on the importance of this work (e.g., climate change, interannual extremes).

Line 31: "the findings," unclear what this is referring to.

Line 32: This paragraph and the next two seem to be organized by snow characteristics and measurement metrics which lead to a statement around line 56 suggesting improvements for snowpack monitoring. It seems that these

three paragraphs could be shortened/combined, while the importance of looking at snow across the entire snow season could be elaborated upon and related back to climate change (or the various drivers of SWE changes, etc.; but with respect to why SwS would shed light on changes in the snowpack moreso than April 1 SWE, for example).

Line 34: Suggest a citation in reference to "and more."

Line 34-35: "the depth of water one would get upon melting a column of snow," "it allows you to quantify the amount of water being stored" – suggest removing "one" and "you" as primary subjects in these sentences, as they read somewhat colloquially and are not necessary for conveying the intended message in each sentence.

Line 35: "reservoir elevation,"… is this meant to say "reservoir at elevation"?

Line 45: Similar comment to above, suggest a citation with "among others" – unclear which metrics are being referred to here.

Line 56: "full time-history," suggest rephrasing (e.g., "a complete evaluation of volumetric and temporal changes in SWE across the entire water year, as opposed to one occurrence in time").

Line 58: "SWE starts the accumulation phase of the snow season up to a peak," suggest rephrasing such that SWE is not "starting" something, but rather (something along the lines of), "SWE accumulation begins and continues until peak SWE occurs, marking the onset of the SWE ablation season."

Additional citations that will need to be included and referenced within the author's introduction of metrics used to evaluate the snowpack (magnitude and timing):
- Hale, K. E., Jennings, K. S., Musselman, K. N., Livneh, B., & Molotch, N. P. (2023). Recent decreases in snow water storage in western North America. *Communications Earth & Environment*, *4*(1).
- Immerzeel, W. W., Lutz, A. F., Andrade, M., Bahl, A., Biemans, H., Bolch, T., ... & Baillie, J. E. M. (2020). Importance and vulnerability of the world's water towers. *Nature*, *577*(7790), 364-369.

Line 94: "(say a SNOTEL site)" another example of somewhat colloquial language, suggest rephrasing or deleting.

Line 94: Units here are very helpful – reiterating the suggestion to include this in the abstract when talking about changes in SwS.

Line 105: "Datasets used for this paper are summarized in Table 1, and briefly reviewed here." Suggest starting this paragraph with the review and ending the paragraph with "a full, comprehensive list of the datasets used in this work are listed in Table 1."

Line 114: "While we recognize the potential limitations of using a modeled SWE product" – this might be addressed later (edit: I do not believe this was addressed later in the discussion), but currently suggest at least listing some of the potential limitations to which the authors refer to here, in addition to citations.

Line 133: "locations that have a mean of at least 30 snow covered days per year based on a 39-year climatology (1982-2021)" – this is determined using the UASWE dataset? If so, please state here.

Line 140: "To answer the first research question, are there significant trends in SwS$_A$ and SwS$_M$ across the US" suggest brevity here (and all similar sentences throughout) by deleting the first clause, "to evaluate significant trends in SwS$_A$ and SwS$_M$ across the US"…

Line 148: "This study used the Hamed and Rao Modified MK test fron the pyMannKendall python package to compute trends in SwS (Hussain and Mahmud, 2019)." Suggest starting this paragraph with this sentence. The paragraph currently leads with describing the MannKendall test and leading the reader to believe that unmodified MannKendall tests were performed. In general, suggest starting the paragraph with the major takeaway/subject and clarifying after the fact.
- Also spelling: "fron"

Header "2.4.2 SwS Trends in Mountain Ecoregions": This paragraph seems to focus primarily on how hypsometry was calculated and how the data in each ecoregion was binned accordingly. Thus, suggest changing this title accordingly.

Line 171: "figure1" add space.

Line 173: "1b", does "figure" need to be added here?

Line 178: "One hundred twenty-three of the stations of the 367 stations with decreasing SwS$_A$ trends, had significant decreases." Suggest being consistent with when numbers are spelled out or not (e.g., 33.5% of the stations with decreasing trends in SwS showed significant decreases (123 of the 367 stations).
- Further, could the authors add the range of significant declines in meter-days? That will further help contextualize the listed percent changes presented in the text and the figures.

Line 179: "The is a mean" grammar.

Line 188: "Remember that the station network only includes the western portion of the US." Suggest deleting.

Line 189: "Only 5% of US grid cells have significant increasing trends, and have a mean percent increase of 84.4%." Delete "," – but also, depicting this 84.4% increase with raw units would again be helpful here. Further, perhaps this is coming, but can the authors share **how** the SwS is increasing (or decreasing)? There are a few sentences that following regarding increases in precipitation but also increased winter temperatures – how has the SWE curve changed in response that has led to such large SwS increases?

Line 199-202: Suggest stating which ER3s the authors are referring to when they point out "only one" or "while four" and the "only one" mountainous ecoregion which shows insignificant increases in SwS.

Line 202: When aggregating the UASWE data to the ER3 scale, is the average annual SwS value an average of its constituents (e.g., is one SWE curve generated per eco-region per year?) or an average of the grid-cell SwS values per year? These values would likely be different and mean different things. I understand that the 93.8% is the division of 15 / 16 ecoregions.

Line 206: Please list number of stations experience a loss in monthly SwS in November where stated.

Paragraph 204-215: Here is another location in the text where it would be fruitful to learn more about how the SWE curve and thus SwS (in raw units) have changed in various locations. In general, the annual and monthly % changes are important to state, however understanding how these changes show up in the SWE curve seems complementary and particularly helpful for the readership in grasping this new metric. Figure 1 does a great job in illustrating why the SwS is important and novel. However, giving examples of annual and monthly SwS values next to actual data and SWE curves used in this analysis seems imperative in communicating these otherwise somewhat abstract % changes.

Line 221: "An average of 72% of the annual SwS$_A$ in the US is held in the 16 mountain ER3s, despite these ER3s only covering 16% of the US land area." Please refer back to Figure where this is shown.

Line 223: "Across all mountain ER3s, there has been a 22% decline in SwS$_A$ over the 39 year period of study" – Same question as above, how are these data averaged across the 16 ER3s?

Line 228-229: These sentences read as though they belong in the introduction or discussion section.

Line 232: How exactly has the SwS$_A$ decreased in most ecoregions and increased in the North Cascades? Generally, how do these changes relate to the examples shown in Figure 1?

Line 238: "highest" – please clarify, highest in elevation?

Line 243: "Looking across all mountain ER3s, there are only significant declining $SwS_A$ trends and no significant increasing trends." It is not clear what led to this statement, which is different from the paragraph above stating that $SwS_A$ increased in the North Cascades – is it related to significance? I'm sure it is so important to continually state trends that are not significant throughout this document – or rather, they should be listed as having no change.

Line 245: "we are able to get an idea"… suggest rephrasing.

Line 247: "This could be a result of increasing snow variability as freezing levels move to higher elevations, resulting in increased irregularity in precipitation form." It would be helpful here again to understand how the SWE curve and thus SwS is changing in these areas – in general, there is a need to couple the stated % changes to the actual data and examples of calculating actual SwS values.

Line 250: "Rige and Valley" spelling.

Paragraph 245-251: Still not quite clear what $SwS_D$ is (not entirely defined at its introduction in line 162) – it might be helpful to reclarify here, however, since this is a section regarding $SwS_A$ trends on the landscape.
- What exactly does "on the landscape" refer to? Is this referring to the binning of SwS values per elevation band? If so, suggest rephrasing title/header.

Line 255: "The percent of stations with negative trends was greater than the percent of stations with positive and positive significant trends in all metrics considered." What exactly does this mean? So, this is a comparison of the # of stations with positive and negative trends per metric?

Line 260: "percent chance" spelling/grammar.

Paragraph 260-269 & Figure 11: This is helpful in beginning to shed light on the utility of the SwS over other metrics. However, without shedding light on the mechanism behind SwS change (how is the SWE curve changing which is causing for changes in the SwS), the changes in SwS relative to other metrics is challenging to contextual and compare/contrast.

Line 271-273: Suggest several citations here regarding the snowpack change literature.

First discussion paragraph: Suggest emphasizing the uniqueness of the SwS compared to other metrics here, in addition to the changes revealed in this work.

General discussion comment: I am less familiar with the required format of this paper, but the equations provided toward the end of the discussion seem out of place – more appropriate in the methods or results section.

Line 293: I believe "outsized" is one word without the hyphen.

Line 295-297: Suggest making the direct connection to SwS and the 13 water basins – e.g., why would SwS be more appropriate here than SWE metrics/other metrics?

Line 315: "Assuming there have not been systematic changes in synoptic weather patterns" – is this a fair assumption? At the very least, a citation should be added here.

Line 320: "Comparison of the various snow metrics provides insight as to how the SWE curve is changing." – From the readership standpoint, I'm not actually sure this is true/has been completed, depending on how one defines "change" here. Added insight has been provided on the magnitude of SwS change compared to other metrics – but changes around the actual shape of the SWE curve and how that relates to changes in SwS have not wholly been described, and I believe that is a critical need in this manuscript.

Line 337: This description or visual explanation of the changing SWE curve is what is desired much earlier in the manuscript – a coupled figure would also be appropriate (ideally showing example changes in SWE curves in the different areas mentioned in this text).

Figure 2: I see that the ER code and name are located in Table 2 – suggest adding that information with this figure if Table 2 will not be in close proximity as in the current manuscript draft.

Figure 10: Units are m$^3$days – this seems inconsistent or at least confusing to the meter-days units explained in the text.

---

## Author Comment (AC1)

We thank the Anonymous Reviewer 2 (AR2) for taking the time to review our manuscript and provide helpful comments. In this document, the black text is ours, the blue text is that of AR2. In the 'line-by-line' section, our response is indented.

General comments:
The study defines a new snow metric, snow water storage (SwS), which is the integrated area under the snow water equivalent (SWE) curve and analyzed the SwS change in America during 1982-2020. It is a good idea to define a new metric (SwS) to express amount of snow water over a time period (water year, a given month, etc.) of interest, and the manuscript is well organized.

Thank you. The need for this new metric was indeed what motivated us to undertake this work.

Specific comments:
(1) The SwS is from SWE integrated over time and SWE itself is the amount of snow accumulated rather than an increase at the corresponding time. The physical meaning of this metric (SwS) doesn't seem to be clear. It seems easier to understand if the SWS is divided by the corresponding time, that is, the average SWE over a period. Moreover, some studies have analyzed the change of mean SWE in some regions over corresponding time periods e.g., Pulliainen et al (2020). Why did you define new metric of SwS instead of average SWE?

This is a fair question, so let us clarify. Things like maximum SWE or 1 April SWE are simple 'snap-shots' that do not give any information about the presence of snow over time. To give an extreme example, imagine a watershed where there was no snow for the entire season except for one brief storm that left 1m of snow that then immediately melted. The idea that 1m of snow (the max SWE) is a good representation of the overall snow season is poor. That is precisely why we integrated the snow over the season. Snow being held back for a long time has ecological benefits (denning of animals; insulation of the ground; etc.) and we wished to create a holistic metric that considers this.

You raise the option of simply using the average SWE as a measure. Again, let's use a hypothetical to make a point. Imagine we have three watersheds. The first has 1m SWE for one week of the year, and zero SWE at other times. The second has 1m SWE for one month of the year, and zero for all other times. The third has 1m SWE for 6 months, and zero for all other times. All three watersheds would report the same average SWE…however, the role that snow is playing in those three watersheds is very different. So, we continue to believe that the integrated (over time) metric plays an important role that is not currently being played by the other useful snow metrics out there.

(2) The snow water storage (SwS) indicated snow mass or average SWE in some studies like Pulliainen et al (2020), Kwon et al 2016,2017, Hale et al 2023, which indicated different meaning in your study. Perhaps you should make a distinction, such as highlighting the meaning in the introduction, or changing the name of your metric.

This is a good comment. First, the two articles by Kwon refer to 'snow water storage' in the title, but there is no 'metric' associated with that phrasing. Instead they use 'snow water storage' to simply refer to snow on the landscape. Second, in the Pulliainen article, which we were aware of, they don't use that phrasing at all. That paper is indeed focused on quantifying snow mass on the landscape, but all of their figures simply give values for Hs or SWE, and they are not integrated quantities over time. Now, Hale's paper makes abundant use of the term 'snow water storage,' but again just as an idea…there is no metric that they define called 'snow water storage.' They do introduce the Snow Storage Index (SSI), but it is a very different concept from ours. A more thorough discussion of the SwS metric and other metrics associated with storage has been added to the introduction and the 'Snow Water Storage' section of the paper.

Technical corrections:

- Line 11: 'snow snow season' might be 'snow season'.

- ○ Thank you for catching this; deleted.
- Line3.1: Might 'SwS trend' be 'SwS change trend'? as well as in Line 176, 204 and other places in the manuscript.
  - ○ Thank you for this suggestion. The language has been updated.
- Line 257: 'The was an 18%'?
  - ○ Changed to 'There was an 18%...'
- Line 257-259: This sentence does not correspond to Table 3.
  - ○ The sentence was updated to reflect the table. 'There was a range of 18 in the percent of stations with negative trends and a range of 14 in the number of stations with positive trends across all metrics.'
- Line 259: Why do not picture?
  - ○ We have chosen to present our results as a combination of figures, tables and statements in order to communicate our results as clearly as possible. Since we cannot include a figure for everything, this is an example of where we decided to use a statement to communicate our results.
- Line 278: There are double 'that'.
  - ○ Fixed.
- Line 280: 'are know known' might be 'are known'.
  - ○ Yes - thank you, this has been fixed.
- Line 311: Harpold et al. (2012) might be (Harpold et al., 2012).
  - ○ Corrected as suggested.
- It might be better to understand if Figure 4 has same extent and scale with Figure 5 and 6.
  - ○ We thank the reviewer for this suggestion, but because we had to balance suggestions from the first reviewer, we decided to modify this figure to show raw SwS change alongside the percent change in SwS. In order to preserve space, we did not change the figure extent.
- Line 182 and 183: There are Northern and Middle Rockies、Southern Rockies and in the Cascades. Can you show their locations in Figure2 although I saw there are mountain names in Table 2 in the later section. The mountains names indicated by numbers should be noted in Figure2 or front of the manuscript.
  - ○ Good suggestion. We updated figure 2 to include ecoregion names.

---

## Author Comment (AC2)

We thank the Anonymous Reviewer 1 (AR1) for taking the time to review our manuscript and provide helpful comments. In this document, the black text is ours, the blue text is that of AR1. In the 'line-by-line' section, our response is indented.

**Review: Changing Snow Water Storage in Natural Snow Reservoirs**

**Summary**

The authors of "Changing Snow Water Storage in Natural Snow Reservoirs" (Aragon and Hill) present a new metric, Snow Water Storage (SwS) to evaluate the snowpack in the mountainous United States (U.S.) throughout the length of the snow season in meter-days. Unlike traditional metrics used to quantify and characterize the snowpack at a single point in time (e.g., April 1 or peak SWE), the SwS captures the area under the SWE curve to illustrate differences and changes in snowpack accumulation and ablation seasons as well, better illustrating the nature of the complete snow season (in a given area). While the metric has great and complementary utility in quantifying the snowpack (on a monthly, annual, or by elevation bin scale), the manuscript would benefit from further depicting the SwS using actual examples across the U.S. in raw units (meter-days as opposed to predominantly reported % changes). Hypothetical examples of the SwS and changes in the SwS are presented in Figure 1 of the manuscript, yet observed and modeled changes in SwS are reported as only % changes. In order to contextualize these changes and further emphasize the added utility of this metric, the readership needs to learn how the SWE curve has changed in various parts of the U.S. to understand why the SwS has increased or decreased, and how the SwS thus provides more/added information compared to other metrics. Translating what is presented in Figure 1 to the real/raw observed and modeled data which are used in the presented work is a critical missing component to this work and would add more intuition around the new metric. Toward the tail end of the discussion, readers learn that "the conceptual SWE curve has been flattening over the 39-year period of record," which is the first mention of how the SWE curve has changed (not just monthly or annual SwS % changes), by way of the SwS evaluation, and thus provides valuable, new information (but also leaves the reader questioning, for example, how is this different from a lower April 1 SWE value? What more does this tell us about the changing snow season?). These questions need to be directly addressed (and seemingly can be, by way of the information gained from SwS). Since the metric leans on the important of temporal changes in the snow season – in addition to changes in magnitude, and thus a novel combination of snowpack characteristics – the changes in SWE curve shape need to be reported throughout the manuscript when % changes are stated (with complementary figures, ideally). This will greatly assist the readership reach the intended conclusions made in the manuscript. The manuscript would benefit from further elaborating on other, recent metrics aimed at quantifying snow water storage (e.g., Hale et al., 2023; Immerzeel et al., 2020) and being more specific in naming changes seen within individual ER3s (instead of "only one" or "four" ER3s, the authors should state the specific areas of reference), such that comparisons can be further made between the SwS and observed changes in other metrics.

The above paragraph gives the overall narrative impression of the paper by AR1, along with some specific suggestions. One of the requests is to present the results (or at least some of them) in terms of raw units, and not just % change. This is a reasonable suggestion since, in areas with little snowpack, a large % change may amount to minimal actual change in SWE. And, we tailored the presentation of our results to address this. Raw change units were added to figures 4, 7, 8 and 9 as well as throughout the text of the paper.

There is a larger, more conceptual, suggestion that we strive to better distinguish the 'value' of SWS changes over, for example, Apr1 SWE changes. In other words, what does changing SWS tell us that changing SWE does not tell us? This is a valuable remark. We attempted to do this in the discussion and with Figure 11 in the original draft, but we will try to strengthen this portion of the paper and we added a more detailed discussion of the SwS metric when it is defined and compared to other snow metrics. Figures 11-13 and their associated discussion also aim to highlight the utility and uniqueness of the SwS metric.

Finally, AR1 urges us to be more complete in our review of recent related efforts, with two specific examples given. It was an oversight to not include the TWI from the Immerzeel et al. paper. And, we were aware of the Hale work, but it was not published at the time of our submission, so we did not include it. Neither of these two other works duplicate what we are doing, and they all have individual, complementary contributions. In our revision we added a good explanation of these other works in order to better demonstrate our specific contribution.

**Line-by-line**

We would like to acknowledge the considerable effort that went into compiling such a detailed list of useful specific edits and suggestions (below). Please see our individual responses.

- Line 7: "An average of 72% of the annual $SwS_A$" – This is a challenging number to capture as a reader. Perhaps defining/clarifying the SwS in terms of units (in the abstract) would help.
    - "On average, the $SwS_A$ in the US is 2.2e6 md." was added to the abstract to give context.

- Line 20: Suggest eliminating "at river headwaters," since snow functions as a reservoir to some degree anywhere in a watershed/landscape.
    - Changed to 'functioning as a natural and spatially-distributed reservoir.'

- Line 10: "The greatest SwSM loss occurs early in the snow snow season, particularly in November" – suggest stating the % loss or magnitude of loss, similarly with the next statement regarding "more spatially widespread significant decreases… than increases" – adding numerical statistics (as either a percent or raw value) would help contextualize the results here.
    - Numerical statistics were added throughout the introduction to help the reader contextualize the results.

- Also, delete "snow" where used twice in a row
    - Thank you for catching this; deleted.

- Was the p-value for determining significance 0.05? If so (or if not), please clarify here.
    - P = 0.1 throughout paper.

- Line 19: "Keystone," since there are many definitions to this term (including political and ecological perspectives, both of which the authors speak to later in this paragraph), suggest using a more precise term that is more synonymous with what is intended – or further defining this term.
    - Changed to 'critical'

- Line 30: Suggest elaborating on why snow water storage may be changing, further shedding light on the importance of this work (e.g., climate change, interannual extremes).
    - This is a reasonable request, and one that is consistent with the main critiques of AR1.

- Line 31: "the findings," unclear what this is referring to.
    - Edited to add clarity: 'This work will evaluate how snow water storage is changing in mountain ecoregions and how these changes may relate to ecosystem and human-related impacts.'

- Line 32: This paragraph and the next two seem to be organized by snow characteristics and measurement metrics which lead to a statement around line 56 suggesting improvements for snowpack monitoring. It seems that these three paragraphs could be shortened/combined, while the importance of looking at snow across the entire snow season could be elaborated upon and related back to climate change (or the various drivers of SWE changes, etc.; but with respect to why SwS would shed light on changes in the snowpack moreso than April 1 SWE, for example).
    - These paragraphs were edited to be more concise. Additional paragraphs were added to discuss additional relevant snow metrics and to discuss various snowpack regimes.
- Line 34: Suggest a citation in reference to "and more."
    - We simply removed the 'and more,' believing the rest of the list to be adequate.

- Line 34-35: "the depth of water one would get upon melting a column of snow," "it allows you to quantify the amount of water being stored" – suggest removing "one" and "you" as primary subjects in these

sentences, as they read somewhat colloquially and are not necessary for conveying the intended message in each sentence.
- We have changed this accordingly.

- Line 35: "reservoir elevation,"… is this meant to say "reservoir at elevation"?
  - We have changed the phrasing of this to be more clear.

- Line 45: Similar comment to above, suggest a citation with "among others" – unclear which metrics are being referred to here.
  - We have removed this phrasing, as the provided list is adequate.

- Line 56: "full time-history," suggest rephrasing (e.g., "a complete evaluation of volumetric and temporal changes in SWE across the entire water year, as opposed to one occurrence in time").
  - We have changed the wording of this somewhat.

- Line 58: "SWE starts the accumulation phase of the snow season up to a peak," suggest rephrasing such that SWE is not "starting" something, but rather (something along the lines of), "SWE accumulation begins and continues until peak SWE occurs, marking the onset of the SWE ablation season."
  - We have rephrased this generally along these lines.

- Additional citations that will need to be included and referenced within the author's introduction of metrics used to evaluate the snowpack (magnitude and timing): Hale, K. E., Jennings, K. S., Musselman, K. N., Livneh, B., & Molotch, N. P. (2023). Recent decreases in snow water storage in western North America. *Communications Earth & Environment*, *4*(1); Immerzeel, W. W., Lutz, A. F., Andrade, M., Bahl, A., Biemans, H., Bolch, T., ... & Baillie, J. E. M. (2020). Importance and vulnerability of the world's water towers. *Nature*, *577*(7790), 364-369.
  - Thank you, these citations and a discussion of their relevance to this paper have been added.

- Line 94: "(say a SNOTEL site)" another example of somewhat colloquial language, suggest rephrasing or deleting.
  - We have simply deleted this, as it was not needed.

- Line 94: Units here are very helpful – reiterating the suggestion to include this in the abstract when talking about changes in SwS.
  - We explicitly added units to the equation and they are discussed in the paragraph below as well.

- Line 105: "Datasets used for this paper are summarized in Table 1, and briefly reviewed here." Suggest starting this paragraph with the review and ending the paragraph with "a full, comprehensive list of the datasets used in this work are listed in Table 1."
  - This is a good suggestion and we have done so.

- Line 114: "While we recognize the potential limitations of using a modeled SWE product" – this might be addressed later (edit: I do not believe this was addressed later in the discussion), but currently suggest at least listing some of the potential limitations to which the authors refer to here, in addition to citations.
  - Limitations of modeled products and citations have been added to this paragraph.

- Line 133: "locations that have a mean of at least 30 snow covered days per year based on a 39-year climatology (1982-2021)" – this is determined using the UASWE dataset? If so, please state here.
  - Correct, and we now state this.

- Line 140: "To answer the first research question, are there significant trends in $SwS_A$ and $SwS_M$ across the US" suggest brevity here (and all similar sentences throughout) by deleting the first clause, "to evaluate significant trends in $SwS_A$ and $SwS_M$ across the US"…

- We have simplified the wording in line with this suggestion.

- Line 148: "This study used the Hamed and Rao Modified MK test fron the pyMannKendall python package to compute trends in SwS (Hussain and Mahmud, 2019)." Suggest starting this paragraph with this sentence. The paragraph currently leads with describing the MannKendall test and leading the reader to believe that unmodified MannKendall tests were performed. In general, suggest starting the paragraph with the major takeaway/subject and clarifying after the fact.
    - Thank you for this feedback, we have adjusted accordingly.

- Also spelling: "fron"
    - Thank you for catching this…fixed.

- Header "2.4.2 SwS Trends in Mountain Ecoregions": This paragraph seems to focus primarily on how hypsometry was calculated and how the data in each ecoregion was binned accordingly. Thus, suggest changing this title accordingly.
    - Upon reflection, we are in agreement and have modified the subsection heading.

- Line 171: "figure1" add space.
    - Done, thank you for catching this.

- Line 173: "1b", does "figure" need to be added here?
    - We now refer to 'panel b' to be more clear.

- Line 178: "One hundred twenty-three of the stations of the 367 stations with decreasing $SwS_A$ trends, had significant decreases." Suggest being consistent with when numbers are spelled out or not (e.g., 33.5% of the stations with decreasing trends in SwS showed significant decreases (123 of the 367 stations). Further, could the authors add the range of significant declines in meter-days? That will further help contextualize the listed percent changes presented in the text and the figures.
    - We agree, the text has been edited for consistency and the range of changes in md have been added.

- Line 179: "The is a mean" grammar.
    - Changed to 'There is a mean…'

- Line 188: "Remember that the station network only includes the western portion of the US." Suggest deleting.
    - Agree. This has been removed.

- Line 189: "Only 5% of US grid cells have significant increasing trends, and have a mean percent increase of 84.4%." Delete "," – but also, depicting this 84.4% increase with raw units would again be helpful here. Further, perhaps this is coming, but can the authors share **how** the SwS is increasing (or decreasing)? There are a few sentences that following regarding increases in precipitation but also increased winter temperatures – how has the SWE curve changed in response that has led to such large SwS increases?
    - We think this comment is asking for dimensional SwS change values to be used alongside percent changes, which we have added throughout the text and in figures 4, 7, 8 and 9.

- Line 199-202: Suggest stating which ER3s the authors are referring to when they point out "only one" or "while four" and the "only one" mountainous ecoregion which shows insignificant increases in SwS.
    - Specific ER3 names were added to this discussion.

- Line 202: When aggregating the UASWE data to the ER3 scale, is the average annual SwS value an average of its constituents (e.g., is one SWE curve generated per eco-region per year?) or an average of the

grid-cell SwS values per year? These values would likely be different and mean different things. I understand that the 93.8% is the division of 15 / 16 ecoregions.

- The following text has been added to the SwS Trends subsection in the methods: "To compute SwS at aggregated ER3 scales, the $SwS_D$ (or SWE) was calculated for each water year in each ER3 to create a single SWE curve. Integrating under this curve provides the $SwS_A$ for each year in units of md. To convert the grid-cell or aggregated SwS to m³-days, multiply the SwS values in md by the area of one grid cell, 16,000,000 m²."

● Line 206: Please list number of stations experience a loss in monthly SwS in November where stated.
- Added.

● Paragraph 204-215: Here is another location in the text where it would be fruitful to learn more about how the SWE curve and thus SwS (in raw units) have changed in various locations. In general, the annual and monthly % changes are important to state, however understanding how these changes show up in the SWE curve seems complementary and particularly helpful for the readership in grasping this new metric. Figure 1 does a great job in illustrating why the SwS is important and novel. However, giving examples of annual and monthly SwS values next to actual data and SWE curves used in this analysis seems imperative in communicating these otherwise somewhat abstract % changes.
- Thank you for raising this point. We amended figures 7-9 to include monthly change in raw units of md and included raw SwS values in this paragraph.

● Line 221: "An average of 72% of the annual $SwS_A$ in the US is held in the 16 mountain ER3s, despite these ER3s only covering 16% of the US land area." Please refer back to Figure where this is shown.
- Done

● Line 223: "Across all mountain ER3s, there has been a 22% decline in $SwS_A$ over the 39 year period of study" – Same question as above, how are these data averaged across the 16 ER3s?
- See above description.

● Line 228-229: These sentences read as though they belong in the introduction or discussion section.
- These sentences were moved.

● Line 232: How exactly has the $SwS_A$ decreased in most ecoregions and increased in the North Cascades? Generally, how do these changes relate to the examples shown in Figure 1?
- Thank you for catching this. This was based on an error in the table - the North Cascades have experienced a 6.67 percent decrease in annual SwS. All values in the table have been re-checked and updated.

● Line 238: "highest" – please clarify, highest in elevation?
- Yes - this is in elevation and has been clarified in the text.

● Line 243: "Looking across all mountain ER3s, there are only significant declining $SwS_A$ trends and no significant increasing trends." It is not clear what led to this statement, which is different from the paragraph above stating that $SwS_A$ increased in the North Cascades – is it related to significance? I'm sure it is so important to continually state trends that are not significant throughout this document – or rather, they should be listed as having no change.
- We have re-written this paragraph to increase clarity. The statement saying there was an increasing trend in the North Cascades was an error that has been amended.

● Line 245: "we are able to get an idea"… suggest rephrasing.
- Yes, we have done so, thank you.

- Line 247: "This could be a result of increasing snow variability as freezing levels move to higher elevations, resulting in increased irregularity in precipitation form." It would be helpful here again to understand how the SWE curve and thus SwS is changing in these areas – in general, there is a need to couple the stated % changes to the actual data and examples of calculating actual SwS values.
  - We have added a description of changes in raw SwS values throughout the paper.

- Line 250: "Rige and Valley" spelling.
  - Corrected, thank you.

- Paragraph 245-251: Still not quite clear what $SwS_D$ is (not entirely defined at its introduction in line 162) – it might be helpful to reclarify here, however, since this is a section regarding $SwS_A$ trends on the landscape.
  - This was defined on line 103 (original draft). $SwS_D$ is the same thing as daily SWE. In order to decrease confusion, WE have changed any mention of $SwS_D$ to simply state 'SWE'.

- What exactly does "on the landscape" refer to? Is this referring to the binning of SwS values per elevation band? If so, suggest rephrasing title/header.
  - Thank you - this heading has been changed to '$SwS_A$ Change Trends in Mountain ER3s'

- Line 255: "The percent of stations with negative trends was greater than the percent of stations with positive and positive significant trends in all metrics considered." What exactly does this mean? So, this is a comparison of the # of stations with positive and negative trends per metric?
  - We agree that the wording in this paragraph was awkward. After reviewing the manuscript, we decided to remove this table and the associated text because it did not substantially contribute to our findings.

- Line 260: "percent chance" spelling/grammar.
  - This has been fixed.

- Paragraph 260-269 & Figure 11: This is helpful in beginning to shed light on the utility of the SwS over other metrics. However, without shedding light on the mechanism behind SwS change (how is the SWE curve changing which is causing for changes in the SwS), the changes in SwS relative to other metrics is challenging to contextual and compare/contrast.
  - We have added analysis to examine cases where snow metrics are changing in different directions at the same locations and added discussion about how the SwS metric has utility for various types of snowpacks. The main goal of this paper is to present the SwS metric, apply it and evaluate how it has changed over recent decades. Understanding the mechanisms that drive change is out of the scope of this paper, but is a great idea for future work.

- Line 271-273: Suggest several citations here regarding the snowpack change literature.
  - Several citations have been added.

- First discussion paragraph: Suggest emphasizing the uniqueness of the SwS compared to other metrics here, in addition to the changes revealed in this work.
  - Thank you for this suggestion, we added a new intro paragraph to the discussion that compares SwS to other metrics and elaborates on why it is a unique metric.

- General discussion comment: I am less familiar with the required format of this paper, but the equations provided toward the end of the discussion seem out of place – more appropriate in the methods or results section.
  - Thank you - the equations have been moved to the results section.

- Line 293: I believe "outsized" is one word without the hyphen.
  - This has been fixed.

- Line 295-297: Suggest making the direct connection to SwS and the 13 water basins – e.g., why would SwS be more appropriate here than SWE metrics/other metrics?
  - Thank you for this suggestion. A second paragraph was added to the discussion to make this point.

- Line 315: "Assuming there have not been systematic changes in synoptic weather patterns" – is this a fair assumption? At the very least, a citation should be added here.
  - We have decided to remove this statement.

- Line 320: "Comparison of the various snow metrics provides insight as to how the SWE curve is changing." – From the readership standpoint, I'm not actually sure this is true/has been completed, depending on how one defines "change" here. Added insight has been provided on the magnitude of SwS change compared to other metrics – but changes around the actual shape of the SWE curve and how that relates to changes in SwS have not wholly been described, and I believe that is a critical need in this manuscript.
  - We expanded on our comparison of SwS to other metrics by examining how snow metrics are (or are not) changing in the same direction (figure 12). We also added a case study to show the utility of the SwS metric when a snowpack regime transitions from a mountain-type to ephemeral.

- Line 337: This description or visual explanation of the changing SWE curve is what is desired much earlier in the manuscript – a coupled figure would also be appropriate (ideally showing example changes in SWE curves in the different areas mentioned in this text).
  - This paragraph was moved to the introduction to highlight the utility of the SwS metric early in the paper.

- Figure 2: I see that the ER code and name are located in Table 2 – suggest adding that information with this figure if Table 2 will not be in close proximity as in the current manuscript draft.
  - Good suggestion - ER codes and names have been added to Figure 2.

- Figure 10: Units are $m^3$days – this seems inconsistent or at least confusing to the meter-days units explained in the text.
  - These units come from computing SwS over an area, as opposed to just at a point. This was explained on line 99 of the original draft.

---

## Referee Report (RR1)

**Review: Changing Snow Water Storage in Natural Snow Reservoirs**

**Summary:** I thank the authors for addressing my comments made during the first round of reviews. My overall summary of the work remains consistent, with additional suggestions made below as line-by-line comments. In general, the additions made to the text and figures greatly enhance the work and the introduction of the Snow Water Storage metric (SwS). My remaining comments are with respect to needed clarification in the text and further representation of SwS in the figures (beyond represented *changes* in SwS). My hope is that addressing this round of feedback will further highlight the importance and applicability of the SwS metric. I look forward to seeing an updated version of the manuscript ready for publication soon.

**Line-by-line**
Line 6: Suggest more specific synonym for "special" (particular?).

Lines 9: Suggest stating the *direction* of change when possible. Annual SwS is stated to have decreased across almost all mountainous ecoregions in line 12-14. Also suggest reporting annual trends separately from monthly trends, as opposed to switching back and forth between sentences. For example, the sentence at lines 12-14 could replace the sentence at line 9. Monthly SwS averages and trends could be reported after annual results. Stating results as non-mountainous ecoregions and mountainous ecoregions (since there is a special focus on these regions) might also increase readability, as that distinction is not clear.

Line 10-11: It is unclear if "in mountainous regions" is (or is not) referring to the 16 mountainous ecoregions here.

Line 15 and Line 19: Is there a mechanism that has been explored to explain why monthly SwS has increased and decreased across the area/across elevation bands? If so, it is suggested these results (or potential mechanisms) be added to the abstract to exemplify how the SwS may become more "valuable." With respect to the use of "valuable," what type of applications are the authors referring to when "valuable" is mentioned? Valuable for predicting snow water resources (referring to "provide information on the natural reservoir function of snowpacks" at lines 16 and 17)? Similarly, is "more valuable" in reference to a comparison to other snowpack metrics or a comparison to its value in the past?

Lines 21-24: Suggest citations.

Lines 29-33: Suggest moving these sentences to the end of the introduction or methods.

Line 34: Suggest a citation.

Line 42: It is unclear what "composite" in quotation marks means (could the quotes be removed?).

Line 48-54: While the figure is extremely helpful in demonstrating the SwS concept, it is unclear what the purpose of this figure is in the context of the introduction. Suggest starting this paragraph with the potential research gap that has been identified by the authors after reviewing the metrics in the previous two paragraphs.

Line 75-76: The reference to Greenland and Antarctica seems irrelevant to this study

Line 77: Similar comment to "more valuable" in Line 19, it is unclear what "less useful" is referring to here – less useful for what exactly?

Suggest including a half or full paragraph in introduction on "future of increased climate variability" (taken from abstract).

Line 78-79: Great set up for this final introductory paragraph. The following sentences before the research questions, however, read as though they belong in the methods section. Instead, the authors could emphasize why and where there is a need for quantifying changes in snow water storage in a new, integrated way.

Line 111: Suggest remaining consistent between "we used" vs. "This study also uses" (e.g., "In this study, we also used…"?).

Line 179: "Typical mountain snowpack" is what was used earlier with respect to "mountain snowpack" here (add "average" or "historical" or "typical"?). A definition of "typical mountain snowpack" vs. "permanent or persistent snowpack" vs. "intermittent or ephemeral snowpack" would be helpful to set the readership up for items in the discussion section.

Section 3.1: My outstanding comment here and from my previous review is with respect to reporting how the SwS has changed in these areas (i.e., the physical changes in SWE curve and SwS representation). Figure 11 is an excellent addition to this manuscript – however readership is still left wondering what SwS actually looks like, numerically, across the region and each individual ecoregion. For now, suggest mentioning that the specific ways in which annual SwS has changed will be shown in section 3.3. This is in addition to the way in which Figure 11 exemplifies the SwS' uniqueness to other snow metrics. See further comments below.

Figure 6: The use of actual/raw SwS values in the text helps contextualize the metric – however this figure only shows percent change. Suggest making this a two-panel plot with average annual SwS across the region and then change through time (current figure).

Section 3.2: Perhaps the ecoregions which are considered mountainous vs. non-mountainous could be listed in the text (as shown in Figure 2) and used as a distinction between Section 3.1 and 3.2 (listed first in Section 2.3). In averaging annual SwS across stations – how many of those stations are in and outside of mountainous ecoregions? It would be very beneficial to include those points in Figure 2 (I see they are shown in Figure 4 but without the ecoregion boundaries). Many of the non-mountainous eco-regions would not be represented by those station results (albeit they are represented via the modeling results). As such, the spatial average results from the station data and the results from the model are very different, and I'm not sure those differences are obviously noted in the text. In general, it is challenging to follow which areas in Figure 2 are represented across sections 3.1 and 3.2. And differences in station results between sections 3.1 and 3.2 are challenging to distinguish, given the areas of interest (i.e., entire CONUS above SCD threshold vs. mountainous ecoregions).

Figure 10: Suggest labeling these panels with the name of each ecoregion.

Figure 11: This is a very helpful addition to the manuscript and is the figure that truly highlights the utility of this metric. Again, suggest labeling these panels with the name of each ecoregion, especially since they are most easily referenced by name in the text. Also suggest letting the y-axis change such that readers can see the changes in SWE curves for all ecoregions (especially 23, 62, 67, 69). Suggest labeling each colored line on one panel – it is challenging to decipher what each line indicates from the figure caption. Does the red labeling indicate a significant decrease or increase in annual SwS (currently says "change")? Those differences in direction should be noted. Finally, it is unclear what "conceptual SWE curve" represents from the actual SWE curve or SwS as represented by the datasets. Is the SwS represented here at all? That is unclear but would be the final, most important piece – to see what SwS actually looks like through time (start of study period vs. end of study period). Perhaps even noting the final numerical SwS calculation for the red dotted line triangle and the red solid line triangle would provide context for this metric.

Line 284: Suggest elaborating on or rephrasing "paint the full picture."

Line 287-288: Citation or figure?

Figure 13: In the text, it is suggested that this figure is a side-by-side comparison of annual SwS, April 1 SWE, max SWE, and snow-covered days – however that is not teased apart in the actual figure. This case study would be extremely valuable if the readership could observe how – perhaps – April 1 SWE have not changed (e.g., it is mentioned that April 1 SWE is often 0 here), SCD has increased, but SwS has decreased. In addition to SWE curves, suggest plotting a subpanel of SCD through time, maximum SWE through time, and then SwS through time with

example curves (e.g., first year on record SWE curve + annual SwS value, final year on record SWE curve + annual SwS value).

Line 308-310: As mentioned above and from the first review, *showing* the SWE curve flattening through time, on average, would be very informative to see and complement the results written in Section 3.1 and shown by ecoregion in Figure 11.

Line 311: "More informative" for what? I ask these repetitive questions because it seems a half sentence is often missing in emphasizing the utility of this metrics over others with respect for water resources. These results seem to hint at incorporating the SwS in a water management scenario.

Line 368: This is a comment likely intended for the methods or results, but the elevation bands (low vs. higher) could generally be defined for each region, since Figure 10 (referencing hypsometry) shows results relative to elevation in each ecoregion.

Discussion: Suggest somewhere in this section to interpret and discuss the implications of the changes seen in Figure 11 and the average "flattening of the SWE curve." Does this indicate that melt is occurring earlier in the year and/or more intermittently throughout the winter? Or is less snow falling throughout the year? Or both? How do these results compare to the metrics mentioned in the introduction? (A good example of this is at line 380 – but this is specific to SWE variability at higher elevations).

---

## Author Response (AR2)

**Response to AR1**

We thank the Anonymous Reviewer 1 (AR1) for taking the time to review our manuscript for a second time and provide helpful comments. In this document, the blue text is ours, the black text is that of AR1.

**Review: Changing Snow Water Storage in Natural Snow Reservoirs**

**Summary:** I thank the authors for addressing my comments made during the first round of reviews. My overall summary of the work remains consistent, with additional suggestions made below as line-by-line comments. In general, the additions made to the text and figures greatly enhance the work and the introduction of the Snow Water Storage metric (SwS). My remaining comments are with respect to needed clarification in the text and further representation of SwS in the figures (beyond represented *changes* in SwS). My hope is that addressing this round of feedback will further highlight the importance and applicability of the SwS metric. I look forward to seeing an updated version of the manuscript ready for publication soon.

The above paragraph summarizes remaining feedback from AR1. Our understanding is that the largest outstanding request from reviewer 1 is to see further representation of SwS. In response to this comment, we have updated figures 5 and 6 to include raw SwS change values. We have also included mean SWE curves (the area under which is the SwS) from the initial and final 20 years of the study period to Figure 11. We feel these changes have improved the manuscript and hope AR1 agrees.

**Line-by-line**

Line 6: Suggest more specific synonym for "special" (particular?).

We changed from special to particular.

Lines 9: Suggest stating the *direction* of change when possible. Annual SwS is stated to have decreased across almost all mountainous ecoregions in line 12-14. Also suggest reporting annual trends separately from monthly trends, as opposed to switching back and forth between sentences. For example, the sentence at lines 12-14 could replace the sentence at line 9. Monthly SwS averages and trends could be reported after annual results. Stating results as non- mountainous ecoregions and mountainous ecoregions (since there is a special focus on these regions) might also increase readability, as that distinction is not clear.

Thank you for this recommendation, the language in the abstract has been modified so that it is more concise and direct. Sentences have also been reordered to talk about $SwS_A$ before talking about $SwS_M$.

Line 10-11: It is unclear if "in mountainous regions" is (or is not) referring to the 16 mountainous ecoregions here.

This sentence was changed to "... in the 16 mountainous ER3s" to clarify.

Line 15 and Line 19: Is there a mechanism that has been explored to explain why monthly SwS has increased and decreased across the area/across elevation bands? If so, it is suggested these results (or potential mechanisms) be added to the abstract to exemplify how the SwS may become more "valuable." With respect to the use of "valuable," what type of applications are the authors referring to when "valuable" is mentioned? Valuable for predicting snow water resources (referring to "provide information on the natural reservoir function of snowpacks" at lines 16 and 17)? Similarly, is "more valuable" in reference to a comparison to other snowpack metrics or a comparison to its value in the past?

Regarding line 15 - the mechanism for differences in SwSM change was not explored in this study. Line 19 was changed to "As we move into a future of increased climate variability and increased variability in mountain snowpacks, spatially and temporally flexible snow metrics such as SwS may become more valuable for monitoring and predicting snow water resources."

Lines 21-24: Suggest citations.

Done.

Lines 29-33: Suggest moving these sentences to the end of the introduction or methods.

These sentences have been moved to the methods section.

Line 34: Suggest a citation.

We feel that this is a well-enough accepted fact that a citation is not required. In order to soften our point, we changed the sentence to "Snow water equivalent (SWE) is a relevant snowpack characteristic for many water resources applications."

Line 42: It is unclear what "composite" in quotation marks means (could the quotes be removed?).

Good point, we removed the quotes.

Line 48-54: While the figure is extremely helpful in demonstrating the SwS concept, it is unclear what the purpose of this figure is in the context of the introduction. Suggest starting this paragraph with the potential research gap that has been identified by the authors after reviewing the metrics in the previous two paragraphs.

With no intention of disrespect, the purpose of this figure is to frame the introduction in a way that demonstrates the SwS concept so that readers can contextualize the idea of SwS in relation to the annual SWE curve and other snow metrics. We did restructure the final introduction paragraph to better frame the research gap in response to this comment.

Line 75-76: The reference to Greenland and Antarctica seems irrelevant to this study

The point of this sentence is to contribute to the discussion of diversity in snowpack regimes by providing an example of a snowpack regime that is different from a mountain snowpack regime.

Line 77: Similar comment to "more valuable" in Line 19, it is unclear what "less useful" is referring to here – less useful for what exactly?

We recognize the reviewer's point and have modified this sentence for clarity: "While metrics such as April 1 SWE, $SWE_{max}$ and SCD do a good job of characterizing mountain snowpacks, they are not as useful at capturing the transient nature of ephemeral snowpacks or the lack of an ablation season on ice sheets."

Suggest including a half or full paragraph in introduction on "future of increased climate variability" (taken from abstract).

Thanks for this recommendation. A brief discussion of climate variability was added to the first introductory paragraph.

Line 78-79: Great set up for this final introductory paragraph. The following sentences before the research questions, however, read as though they belong in the methods section. Instead, the authors could emphasize why and where there is a need for quantifying changes in snow water storage in a new, integrated way.

Thank you, we have reworked the final introductory paragraph to better frame the research gap and purpose of this paper.

Line 111: Suggest remaining consistent between "we used" vs. "This study also uses" (e.g., "In this study, we also used..."?).

Thank you for this suggestion, "we" statements have been removed from our methodology to add consistency.

Line 179: "Typical mountain snowpack" is what was used earlier with respect to "mountain snowpack" here (add "average" or "historical" or "typical"?). A definition of "typical mountain snowpack" vs. "permanent or persistent snowpack" vs. "intermittent or ephemeral snowpack" would be helpful to set the readership up for items in the discussion section.

This is a great point. We checked the full text to consolidate our terminology so now we only reference mountain snowpacks, and ephemeral snowpacks. We added wording to the 6th introductory paragraph to clarify what is meant by a mountain snowpack and an ephemeral snowpack in response to this feedback.

Section 3.1: My outstanding comment here and from my previous review is with respect to reporting how the SwS has changed in these areas (i.e., the physical changes in SWE curve and SwS representation). Figure 11 is an excellent addition to this manuscript – however readership is still left wondering what SwS actually looks like, numerically, across the region and each individual ecoregion. For now, suggest mentioning that the specific ways in which annual

SwS has changed will be shown in section 3.3. This is in addition to the way in which Figure 11 exemplifies the SwS' uniqueness to other snow metrics. See further comments below.

Good recommendation, a sentence was added at the end of section 3.1 saying: "The specific ways in which SwS$_A$ has changed across ER3s and how these changes relate to other snow metrics will be discussed in section 3.3." We also added a response to your Figure 11 comments below (page 8).

Figure 6: The use of actual/raw SwS values in the text helps contextualize the metric – however this figure only shows percent change. Suggest making this a two-panel plot with average annual SwS across the region and then change through time (current figure).

We added a subplot to figures 5 and 6 to show the raw SwS values in addition to the percent changes.

[Figure]

Updated Figure 5.

[Figure]

Section 3.2: Perhaps the ecoregions which are considered mountainous vs. non-mountainous could be listed in the text (as shown in Figure 2) and used as a distinction between Section 3.1 and 3.2 (listed first in Section 2.3). In averaging annual SwS across stations – how many of those stations are in and outside of mountainous ecoregions? It would be very beneficial to include those points in Figure 2 (I see they are shown in Figure 4 but without the ecoregion

boundaries). Many of the non-mountainous eco-regions would not be represented by those station results (albeit they are represented via the modeling results). As such, the spatial average results from the station data and the results from the model are very different, and I'm not sure those differences are obviously noted in the text. In general, it is challenging to follow which areas in Figure 2 are represented across sections 3.1 and 3.2. And differences in station results between sections 3.1 and 3.2 are challenging to distinguish, given the areas of interest (i.e., entire CONUS above SCD threshold vs. mountainous ecoregions).

Since there are so many non-mountainous ER3s and they are not the focus of our results, we do not think it is necessary to list them by name in the text. We agree that the distinction between sections 3.1 and 3.3 should be improved. Section 3.1 now covers the non-mountainous ER3s and the discussion about $SwS_A$ changes in mountainous ER3s was moved to Section 3.3.

Annual SwS was not averaged across stations, instead it was calculated at each station individually in the station analysis. In the ER3 analysis, annual SwS was computed at each grid-cell of the UA SWE product and then aggregated to the ER3 scale (see methods section 2.4.1). Because of this, we do not feel we would glean any additional information by evaluating which stations fall into ER3s.

Figure 10: Suggest labeling these panels with the name of each ecoregion.

The figure was crowded with the full ecoregion names, but we did add a legend to facilitate figure interpretation and added "Refer to Table 2 for ER3 names" to the figure caption

to let readers know where they can find ER3 names.

[Figure]

Updated Figure 10.

Figure 11: This is a very helpful addition to the manuscript and is the figure that truly highlights the utility of this metric. Again, suggest labeling these panels with the name of each ecoregion, especially since they are most easily referenced by name in the text. Also suggest letting the y-axis change such that readers can see the changes in SWE curves for all ecoregions (especially 23, 62, 67, 69). Suggest labeling each colored line on one panel – it is challenging to decipher what each line indicates from the figure caption. Does the red labeling indicate a significant decrease or increase in annual SwS (currently says "change")? Those differences in

direction should be noted. Finally, it is unclear what "conceptual SWE curve" represents from the actual SWE curve or SwS as represented by the datasets. Is the SwS represented here at all? That is unclear but would be the final, most important piece – to see what SwS actually looks like through time (start of study period vs. end of study period). Perhaps even noting the final numerical SwS calculation for the red dotted line triangle and the red solid line triangle would provide context for this metric.

We are pleased that the reviewer found Figure 11 to be a useful addition to the paper and we appreciate the additional feedback. Similar to our comment for Figure 10, we found the figure to be overly crowded with the full ecoregion names, so we left their number code instead and added "Refer to Table 2 for ER3 names" to the figure caption to let readers know where they can find ER3 names. We did allow the y-axis to scale with each ER3 as the reviewer suggested and added a legend to facilitate figure interpretation. The reviewer has a good point about indicating the direction of change and the language has been adjusted to specify the direction of change. The language in the introduction was adjusted to provide a description of the conceptual SWE curve as follows:

" The conceptual SWE curve for mountain snowpacks is defined by three points; a DSO the peak SWE ($SWE_{max}$) and the DSD. SWE accumulation begins at the DSO and continues up to a $SWE_{max}$, which may or may not occur on Apr 1 (northern hemisphere). After $SWE_{max}$, the ablation phase of the snow season begins and the SWE depth declines until it reaches zero at the DSD."

We hope this description of the conceptual SWE curve adds clarity for the reviewer. Since SwS is the integrated area under the SWE curve, it is reflected in this figure as the shape below the SWE curve. In response to the reviewer comment, we replaced the mean SWE curve for the full time period with two mean SWE curves, one from the 20 years of the study period and one from the final 20 years of the study period. In this way, one can get a general idea of how the actual SWE curve and associated SwS has changed over time.

[Figure]

Updated Figure 11.

Line 284: Suggest elaborating on or rephrasing "paint the full picture."

This sentence has been rephased and now reads: "The inability of common one-dimensional snow metrics to reflect snow storage change is particularly apparent when snowpacks transition from one snow regime to another, such as a permanent snowpack transitioning to a mountain snowpack or a mountain snowpack transitioning to an ephemeral snowpack."

Line 287-288: Citation or figure?

This statement is based on an analysis that we performed. The sentence was updated to say "This study found the number of annual snow-free periods…" to clarify that this was a result we

found. The figure associated with this result can be found below, though we do not feel this figure needs to be added to the paper.

[Figure]

Figure 13: In the text, it is suggested that this figure is a side-by-side comparison of annual SwS, April 1 SWE, max SWE, and snow-covered days – however that is not teased apart in the actual figure. This case study would be extremely valuable if the readership could observe how – perhaps – April 1 SWE have not changed (e.g., it is mentioned that April 1 SWE is often 0 here), SCD has increased, but SwS has decreased. In addition to SWE curves, suggest plotting a subpanel of SCD through time, maximum SWE through time, and then SwS through time with example curves (e.g., first year on record SWE curve + annual SwS value, final year on record SWE curve + annual SwS value).

Thank you for suggesting this. In our analysis we created plots to explore the trends in snow metrics over time and we agree that the addition of subplots would improve this figure. See the updated Figure 13 below.

[Figure]

Line 308-310: As mentioned above and from the first review, *showing* the SWE curve flattening through time, on average, would be very informative to see and complement the results written in Section 3.1 and shown by ecoregion in Figure 11.

The result of the SWE curve flattening was based on station data, and the relative percent changes were not evaluated at ER3 scales. Additionally, this result is based on a regression analysis that shows that the percent change in SWEmax is greater than would be expected relative to the percent change in SwSA and the percent change in SCD is less than would be expected relative to the percent change in $SwS_A$. We found this result by doing a regression analysis on the 465 available stations, so there is notable variability across stations. If we understand your comment correctly, you are interested in seeing an example of a SWE curve flattening over time for a specific location. We do not feel there is utility in showing SWE curves from a single station because of the variability that exists across stations. Instead, we think that the current regression plots are the most effective means of showing this result.

Line 311: "More informative" for what? I ask these repetitive questions because it seems a half sentence is often missing in emphasizing the utility of this metrics over others with respect for water resources. These results seem to hint at incorporating the SwS in a water management scenario.

Thank you. The sentence was updated to "So, relying on a single metric like April 1st SWE gives an incomplete assessment of the storage of snow throughout a full season, and a more holistic metric like SwS may be more informative when considering a full snow season."

Line 368: This is a comment likely intended for the methods or results, but the elevation bands (low vs. higher) could generally be defined for each region, since Figure 10 (referencing hypsometry) shows results relative to elevation in each ecoregion.

Figure 10 includes elevation, but in a normalized sense. We chose this normalized approach purposefully. If we interpret your comment above correctly, you seem interested in knowing dimensional information (actual elevations) about the ER3s. As noted in the manuscript, 10 elevation bands are defined for each ER3. To accommodate your interest in the raw elevations, we have added a column to Table 2 that provides (min, max) the minimum and maximum elevation for each ER3. With this information, it is straightforward for the reader to understand what the dimensional elevation bands are for each ER3. We retain the normalized presentation in Figure 10, however, so that the reader can easily see what is going on at 'relative' elevations across the different ER3s. Thank you for your comment and we hope that we have read it correctly.

Discussion: Suggest somewhere in this section to interpret and discuss the implications of the changes seen in Figure 11 and the average "flattening of the SWE curve." Does this indicate that melt is occurring earlier in the year and/or more intermittently throughout the winter? Or is less snow falling throughout the year? Or both? How do these results compare to the metrics mentioned in the introduction? (A good example of this is at line 380 – but this is specific to SWE variability at higher elevations).

The second paragraph of the discussion addresses why the findings in Figure 11 are important, but we realized that we did not explicitly reference Figure 11 in our previous submission. We have added a reference to Figure 11 in this second paragraph. As far as the flattening of the SWE curve, this paper is not focused on the mechanism of cause, it is focused on presenting a new snow metric and calculating the history of snow water storage. We added the following to paragraph 3 of the discussion to further interpret our results as they relate to plausible mechanistic drivers of change:

"While future work could explore the exact mechanistic drivers of predominantly decreasing SwS trends, these findings are reasonable in the context of mechanistic drivers explored in other snow change literature. From an energy budget standpoint, snow falling at warmer temperatures (as a result of climate warming) and overall shallower snowpacks (due to reduced snowfall fractions) contribute to reduced cold content and more readily ripening snowpacks

(Jennings and Molotoch, 2020) . Additionally, shallower snowpacks are susceptible to enhanced snowmelt from the albedo feedback as vegetation and soil are exposed (Kapnick and Hall, 2010)."